# Oxidized low-density lipoprotein potentiates angiotensin II-induced Gq activation through the AT1-LOX1 receptor complex

Jittoku Ihara[1†], Yibin Huang[1,2†], Yoichi Takami[1*], Yoichi Nozato[1*], Toshimasa Takahashi[1,3*], Akemi Kakino[4], Cheng Wang[1], Ziwei Wang[1], Yu Guo[1], Weidong Liu[1], Nanxiang Yin[1], Ryoichi Ohara[1], Taku Fujimoto[1], Shino Yoshida[1], Kazuhiro Hongyo[1], Hiroshi Koriyama[1], Hiroshi Akasaka[1], Hikari Takeshita[1], Shinsuke Sakai[5], Kazunori Inoue[5], Yoshitaka Isaka[5], Hiromi Rakugi[1], Tatsuya Sawamura[4], Koichi Yamamoto[1]

[1]Department of Geriatric and General Medicine, Osaka University Graduate School of Medicine, Osaka, Japan; [2]Center for Pulmonary and Vascular Biology, Department of Pediatrics, University of Texas Southwestern Medical Center, Dallas, United States; [3]Department of Medicine, University of Toronto, Toronto, Canada; [4]Department of Molecular Pathophysiology, Shinshu University Graduate School of Medicine, Matsumoto, Japan; [5]Department of Nephrology, Osaka University Graduate School of Medicine, Osaka, Japan

**\*For correspondence:**
takami@geriat.med.osaka-u.ac.jp (YT);
yoichi.nozato@geriat.med.osaka-u.ac.jp (YN);
tkhstsms@hotmail.co.jp (TT)

[†]These authors contributed equally to this work

## eLife Assessment

This study provides **useful** in vitro evidence to support a mechanism whereby dyslipidemia could accelerate renal functional decline through the activation of the AT1R/LOX1 complex by oxLDL and AngII. As such, it improves the knowledge regarding the complex interplay between dyslipidemia and renal disease and provides a **solid** basis for the discovery of novel therapeutic strategies for patients with lipid disorders. The methods, data, and analyses partly support the presented findings, although the observed variability and need for further in vivo validation require additional research in this key area.

**Abstract** Chronic kidney disease (CKD) and atherosclerotic heart disease, frequently associated with dyslipidemia and hypertension, represent significant health concerns. We investigated the interplay among these conditions, focusing on the role of oxidized low-density lipoprotein (oxLDL) and angiotensin II (Ang II) in renal injury via G protein αq subunit (Gq) signaling. We hypothesized that oxLDL enhances Ang II-induced Gq signaling via the AT1 (Ang II type 1 receptor)-LOX1 (lectin-like oxLDL receptor) complex. Based on CHO and renal cell model experiments, oxLDL alone did not activate Gq signaling. However, when combined with Ang II, it significantly potentiated Gq-mediated inositol phosphate 1 production and calcium influx in cells expressing both LOX-1 and AT1 but not in AT1-expressing cells. This suggests a critical synergistic interaction between oxLDL and Ang II in the AT1-LOX1 complex. Conformational studies using AT1 biosensors have indicated a unique receptor conformational change due to the oxLDL-Ang II combination. In vivo, wild-type mice fed a high-fat diet with Ang II infusion presented exacerbated renal dysfunction, whereas LOX-1 knockout mice did not, underscoring the pathophysiological relevance of the AT1-LOX1 interaction in renal damage. These findings highlight a novel mechanism of renal dysfunction in CKD driven

by dyslipidemia and hypertension and suggest the therapeutic potential of AT1-LOX1 receptor complex in patients with these comorbidities.

## Introduction

Dyslipidemia is a major risk factor of atherosclerotic heart disease in patients with CKD. It has been postulated that the association between dyslipidemia and CKD is not solely a result of epidemiological comorbidities but rather a complex interplay of causality, where CKD exacerbates dyslipidemia, while dyslipidemia, in turn, contributes to the onset and progression of CKD (*Harper and Jacobson, 2008*; *Cases and Coll, 2005*). Since Moorhead et al. proposed the lipid nephrotoxicity hypothesis in 1982 (*Moorhead et al., 1982*), accumulating evidence has suggested that increased plasma lipid levels contribute to the development of renal glomerular and tubular damage, primarily in animal models of dyslipidemia (*Ruan et al., 2009*). The etiology of dyslipidemia-induced nephrotoxicity is complex and multifaceted, potentially involving the activation of certain cellular signaling pathways that culminate in renal injury through elevated levels of oxLDL (*Bussolati et al., 2005*; *Dai et al., 2014*; *Deng et al., 2016*; *Gai et al., 2019*). The lectin-like oxLDL receptor, LOX-1, is implicated in organ damage caused by dyslipidemia, and its expression is increased in hypertensive glomerulosclerosis (*Nagase et al., 2000*). In murine models, a deficiency of LOX-1 leads to a reduction in renal dysfunction, which was precipitated by a systemic inflammatory state following significant myocardial ischemia and injury (*Lu et al., 2012*). This supports the hypothesis that LOX-1 plays a role in the development of inflammation-induced renal injury. However, to date, no studies have investigated the role of LOX-1 in nephrotoxicity caused by dyslipidemia.

Hypertension is an established risk factor for CKD, and it has been suggested that hypertension and dyslipidemia act synergistically to induce renal dysfunction (*Mänttäri et al., 1995*). The development of renal damage due to hypertension involves direct renal injury by vasoactive hormones, such as Ang II in addition to renal hemodynamic abnormalities associated with elevated body pressure (*Brewster and Perazella, 2004*). We have shown that LOX-1 and the Ang II type 1 receptor (AT1) of the G protein-coupled receptor (GPCR) are coupled on the plasma membrane, and that G protein-dependent and β-arrestin-dependent AT1 activation mechanisms are involved in the signaling mechanism by oxLDL and the intracellular uptake of oxLDL, respectively (*Yamamoto et al., 2015*; *Takahashi et al., 2021*).

Interestingly, it was recently demonstrated that AT1 exhibits different modes and degrees of G protein activation depending on the conformational changes that occur during activation by various ligands (*Wingler et al., 2019*). Specifically, G protein αq subunit (Gq)-biased agonists induce a more open conformation of AT1 than Ang II, resulting in a more potent activation of Gq, which is the primary mediator of Ang II-induced hypertension. In contrast, β-arrestin-biased agonists induce a closer conformational change, leading to the activation of β-arrestin without the activation of Gq (*Wingler et al., 2019*). Indeed, we found that oxLDL selectively activates G protein αi subunit (Gi) of AT1, without activating Gq (*Takahashi et al., 2021*), similar to that induced by β-arrestin-biased agonists (*Namkung et al., 2018*). However, given that Ang II and oxLDL coexist in physiological environments, it is plausible to hypothesize that the effect of these ligands on AT1 in living organisms may differ from their effects when administered individually. Indeed, we found that the combination treatment of oxLDL and Ang II enhanced pro-inflammatory NFκB activity compared to each treatment alone in cells overexpressing both AT1 and LOX-1 (*Takahashi et al., 2021*). Based on our findings and the aforementioned structure-activity relationship of AT1, we hypothesized that the binding of both oxLDL and Ang II to LOX-1 and AT1, respectively, may result in a more open AT1 structure, leading to stronger downstream Gq signaling. Therefore, this study aimed to investigate and clarify this hypothesis, focusing specifically on renal component cells, and ultimately demonstrate the in vivo relevance of this phenomenon in the development of renal injury or renal dysfunction under conditions of Ang II and oxLDL overload.

## Results

### Oxidized LDL potentiates Ang II-induced Gq signaling in a LOX-1-dependent manner

First, we examined the additive effect of oxLDL on Ang II-stimulated AT-1-Gq signaling in CHO (Chinese hamster ovary) cells that did not endogenously express LOX-1 and AT-1 but were genetically engineered to express these receptors (CHO-LOX-1-AT1) *Yamamoto et al., 2015*.

*Figure 1a* shows the dose-response curve of Ang II-induced IP1 production, which serves as a measure of Gq activity, in the presence of varying concentrations of oxLDL in CHO-LOX-1-AT1 cells. Consistent with previous findings, oxLDL alone did not stimulate IP1 production in CHO-LOX-1-AT1 cells (*Takahashi et al., 2021*). However, when oxLDL was supplemented at concentrations of 10 and 20 µg/mL (but not at 5 µg/mL), it caused a similar leftward shift in the dose-response curve of Ang II, resulting in a more than 80% decrease in the Effective Concentration 50 (EC50) (EC50 values: 0 µg/mL, $9.40 \times 10^{-9}$ M; 5 µg/mL, $5.21 \times 10^{-9}$ M; 10 µg/mL, $1.68 \times 10^{-9}$ M; 20 µg/mL, $1.55 \times 10^{-9}$ M). The maximum IP1 production induced by Ang II in CHO-LOX-1-AT1 cells was unaffected by the addition of oxLDL. Additionally, we observed that oxLDL administration decreased Ang II-induced IP1 production in CHO cells expressing AT1 alone (CHO-AT1), in contrast to CHO cells expressing both AT1 and LOX-1 (*Figure 1b*). This finding suggests that the potentiating effect of oxLDL on Ang II-AT1-Gq activity is dependent on the presence of LOX-1. However, the reason for the decrease in IP1 production caused by oxLDL supplementation was not determined in this study. Native LDL, which does not bind to LOX-1, did not alter Ang II-induced Gq activity in CHO-LOX-1-AT1 cells (*Figure 1c*). Furthermore, the presence of advanced glycation end products (AGEs) that bind to LOX-1 (*Jono et al., 2002*), did not enhance Ang II-induced IP1 production (*Figure 1d*).

We found that enhanced IP1 production by co-treatment of oxLDL with Ang II was similarly observed in CHO-cells expressing LOX-1 and mutated AT1 with impaired ability to activate β arrestin (*Takahashi et al., 2021*; *Figure 1e*). Moreover, the potentiating effect of oxLDL on IP1 production was unaffected by Pertussis Toxin (PTX), a Gi inhibitor, or RKI-1448, a downstream Rho kinase inhibitor targeting G12/13 signaling in CHO-LOX-1-AT1 cells. However, this phenomenon was completely inhibited by YM-254890, the Gq inhibitor (*Figure 1f*). These results suggest that the potentiating effect of oxLDL on Ang II-induced IP1 production is not influenced by β-arrestin, Gi, or G12/13, which are the main effectors of AT1 signaling aside from Gq (*Lymperopoulos et al., 2021*).

### oxLDL potentiates Ang II-induced calcium influx in a LOX-1-dependent manner

Calcium influx is a representative cellular phenomenon that occurs in response to Ang II-AT1-Gq activation. We found that oxLDL and low concentrations of Ang II ($10^{-12}$ M) did not induce calcium influx in CHO-LOX-1-AT1 cells when treated alone, but apparently, induced calcium influx when treated together (*Figure 1g and h*). Ang II alone at higher concentrations ($10^{-11}$ M) induced calcium influx, and no further enhancement was observed with oxLDL supplementation (*Figure 1h*). The combined effect of oxLDL and Ang II on calcium influx was completely blocked by YM254890, suggesting that this phenomenon was Gq-dependent (*Figure 1i*). Importantly, oxLDL supplementation with Ang II did not affect the calcium influx in CHO cells expressing AT1 alone (*Figure 1j*).

### Co-treatment of oxLDL with Ang II induces conformational change of AT1 different from each treatment alone

To gain mechanistic insight into this phenomenon, we initially conducted a live-imaging analysis of membrane LOX-1 and AT1 to determine whether the combined treatment of oxLDL and Ang II leads to increased internalization of AT1 upon activation compared to individual treatments, as described in a previous protocol paper (*Huang et al., 2022*; *Figure 2—figure supplement 1*, *Video 1*, and *Figure 2a*).

Our findings revealed a decrease in green puncta, which represent AT1-eGFP, upon treatment with Ang II, oxLDL alone, and their co-treatment compared with the control group (*Figure 2a*). However, the extent of membrane AT1 reduction was similar across all the treatment groups (*Figure 2a*). Consistent with our previous report (*Takahashi et al., 2021*), oxLDL treatment resulted in a reduction in the red puncta representing LOX-1-mScarlet (*Figure 2a*). Importantly, co-treatment with oxLDL and Ang II

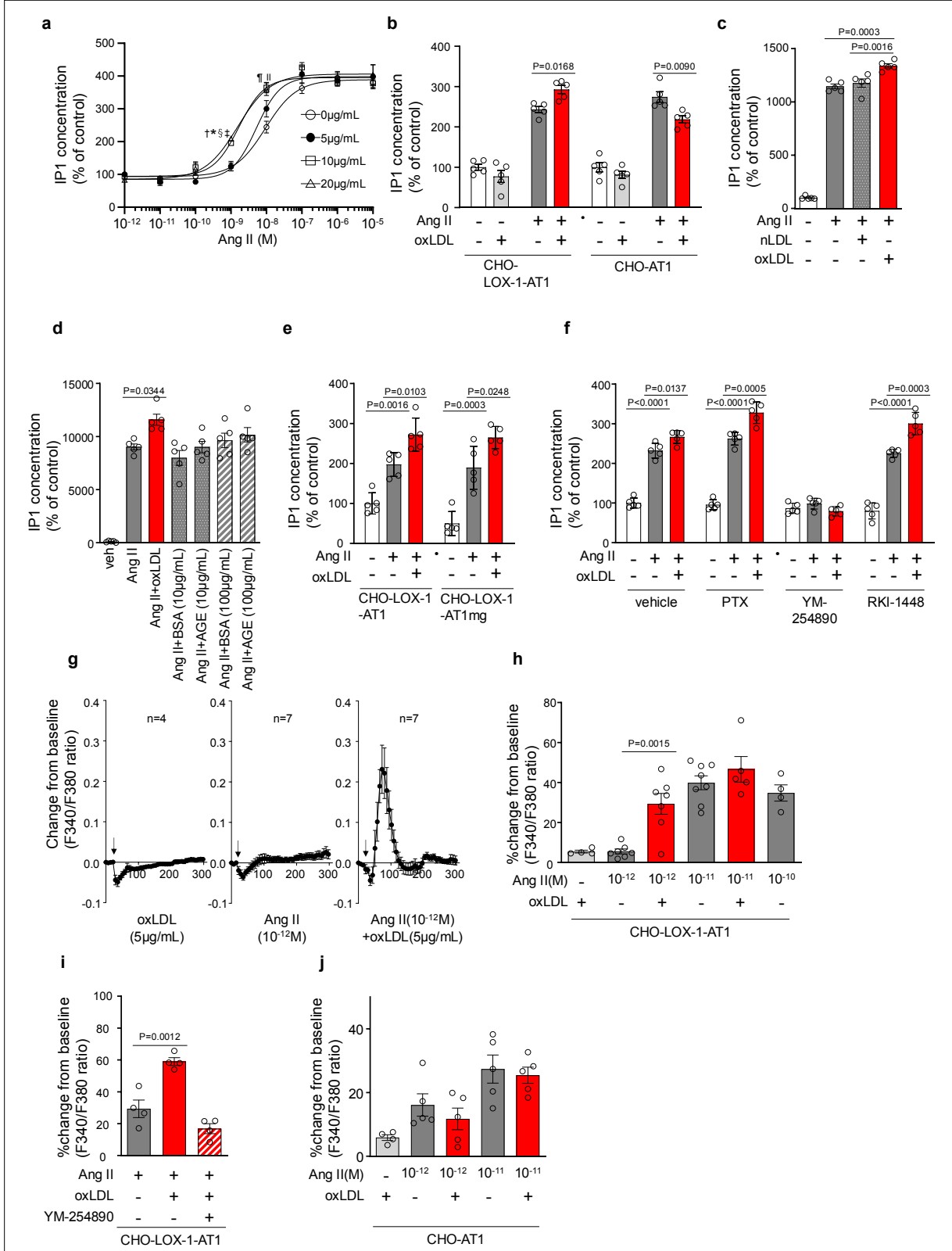

**Figure 1.** Oxidized low-density lipoprotein (LDL) potentiates angiotensin II (Ang II)-induced G protein αq subunit (Gq) signaling and calcium influx in a LOX-1-dependent manner. (**a**) Dose-dependent response of IP1 concentration by the activation of Gq signaling in response to oxLDL and Ang II in CHO-LOX-1-AT1 cells. Cells were treated with oxLDL and Ang II at the concentrations described in the Figure (n=4 for each oxLDL concentration). *p=0.0004 for 0 µg/mL vs 20 µg/mL, †p=0.0003 for 5 g/mL vs 20 µg/mL, ‡p=0.0020 for 0 µg/mL vs 10 µg/mL, and §p=0.0015 for 5 µg/mL vs 10 µg/mL at

*Figure 1 continued on next page*

Figure 1 continued

$10^{-9}$ M Ang II; $^{\parallel}$p=0.0051 for 0 µg/mL vs 10 µg/mL, $^{\P}$p=0.0004 for 0 µg/mL vs 20 µg/mL at $10^{-8}$ M Ang II. Biological replicates were performed using two independent cell cultures. (**b**) IP1 concentration in response to vehicle, native LDL (nLDL 10 µg/mL), and oxLDL (10 µg/mL) in the combination of Ang II ($10^{-8}$ M) in CHO-LOX-1-AT1 cells (n=5 for each group) biological replicates were performed using two independent cell cultures. (**c**) IP1 concentration in response to vehicle, oxLDL (10 µg/mL) in the combination of Ang II ($10^{-8}$M) in CHO-LOX-1-AT1 and CHO-AT1 cells (n=5 for each group). (**d**) IP1 concentration in response to vehicle, oxLDL (10 µg/mL), BSA (10 or 100 µg/mL), BSA-conjugated AGE (10 or 100 µg/mL) in the combination of Ang II ($10^{-8}$ M) in CHO-LOX-1-AT1 cells. (**e**) IP1 concentration in response to vehicle, oxLDL (10 µg/mL) in the combination of Ang II ($10^{-8}$ M) in genetically engineered Chinese hamster ovary (CHO) cells with or without intact β-arrestin binding domain (n=5 for each group). AT1mg indicates AT1 a mutant AT1 lacking a functional β-arrestin binding domain but retaining G-protein-biased signaling capability. (**f**) IP1 concentration in response to oxLDL (10 µg/mL) in the combination of Ang II ($10^{-8}$ M) and additional effect of PTX, a Gi inhibitor, YM-254890, a Gq inhibitor, and RKI-1448, a downstream Rho kinase inhibitor targeting G12/13 signaling, in CHO-LOX-1-AT1 cells (n=5 for each group). (**g**) Intracellular calcium dynamics measured using Fura 2-AM by the ratio of the mission signals at excitation wavelength 340 nm and 380 nm in response to oxLDL (5 µg/mL), Ang II ($10^{-12}$ M), and their combination (for each agonist, 4–7 regions of interest were selected). Addition of these agonists is marked with arrows on the timeline of the assay. (**h**) Percentage changes from baseline in the ratio of emission signals (F340/F380) measured by Fura 2-AM were quantified following treatment with oxLDL and Ang II at specified concentrations in CHO-LOX-1-AT1 cells as detailed in the figure (n=4–8). (**i**) Percentage change from baseline in the ratio of emission signals (F340/F380) measured by Fura 2-AM after stimulation with oxLDL (5 µg/mL), Ang II ($10^{-12}$ M), and YM-254890, a Gq inhibitor, in CHO-LOX-1-AT1 cells. (**j**) Percentage changes from baseline in the ratio of emission signals (F340/F380) measured by Fura 2-AM were quantified following treatment with oxLDL (5 µg/mL) and Ang II at specified concentrations in CHO-AT1 cells, as detailed in the Figure Biological replicates were performed using two independent cell cultures. Data are represented as mean ± SEM. Differences were determined using one-way ANOVA, followed by Tukey's multiple comparison test for (**a-f**) and (**h-j**).

The online version of this article includes the following source data for figure 1:

**Source data 1.** Oxidized low-density lipoprotein (LDL) potentiates angiotensin II (Ang II)-induced G protein αq subunit (Gq) signaling and calcium influx in a LOX-1-dependent manner.

did not further enhance the reduction of red puncta compared to oxLDL treatment alone (*Figure 2a*). Based on these findings, it is conceivable that co-treatment with oxLDL and Ang II does not increase the content of activated AT1 compared to individual treatments alone.

AT1 intramolecular FlAsH-BRET biosensors were used to detect AT1 conformational changes in CHO cells expressing LOX-1 (*Devost et al., 2017*). Among several biosensors previously tested (*Devost et al., 2017*), we used two sensors with FlAsH insertion at the third intracellular loop (ICL3P3) and cytoplasmic-terminal tail (C-tailP1) of AT1 that interacts with Renilla luciferase (RlucII) at the end of the cytoplasmic tail (AT1-ICL3P3 and AT1-C-tailP1, respectively), as these sensors were shown to enhance BRET signaling induced by Ang II compared to biased agonists, including SI, SII, DVG, and SBpA (*Devost et al., 2017*). In CHO cells expressing LOX-1 alone (CHO-LOX-1) transduced with lentivirus encoding AT1-C-tailP1, $10^{-5}$ M Ang II and the combination of Ang II and 10 µg/mL oxLDL induced BRET similarly, whereas oxLDL alone did not alter BRET (*Figure 2b*). In contrast, in CHO-LOX-1 cells transduced with AT1-ICL3P3-encoded lentivirus, the combination of Ang II and oxLDL induced BRET more prominently than Ang II alone, whereas oxLDL alone did not alter BRET (*Figure 2c*). The difference in sensitivity between the CHO-LOX-1-AT1-3p3 and CHO-LOX-1-AT1-C-tail P1 sensors likely explains why only the former showed a significant response to the combination of Ang II and oxLDL, underscoring the importance of FlAsH insertion site selection in these assays. Furthermore, the difference between oxLDL and the combination treatment was abolished by a neutralizing antibody against LOX-1 (*Figure 2d*). These findings suggested that

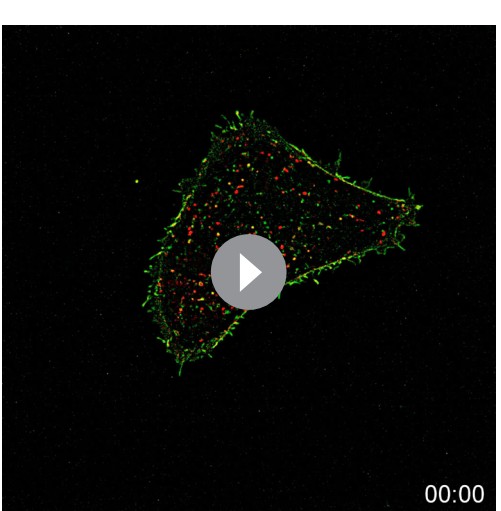

**Video 1.** Live-imaging of membrane LOX-1 and AT1 in response to the co-treatment of oxidized low-density lipoprotein (oxLDL) with AngII. Real-time membrane imaging of Chinese hamster ovary (CHO) cells co-transfected with LOX-1-mScarlet and AT1-eGFP in response to oxLDL (10 µg/ml) in the combination of angiotensin II (Ang II) ($10^{-7}$M).

https://elifesciences.org/articles/98766/figures#video1

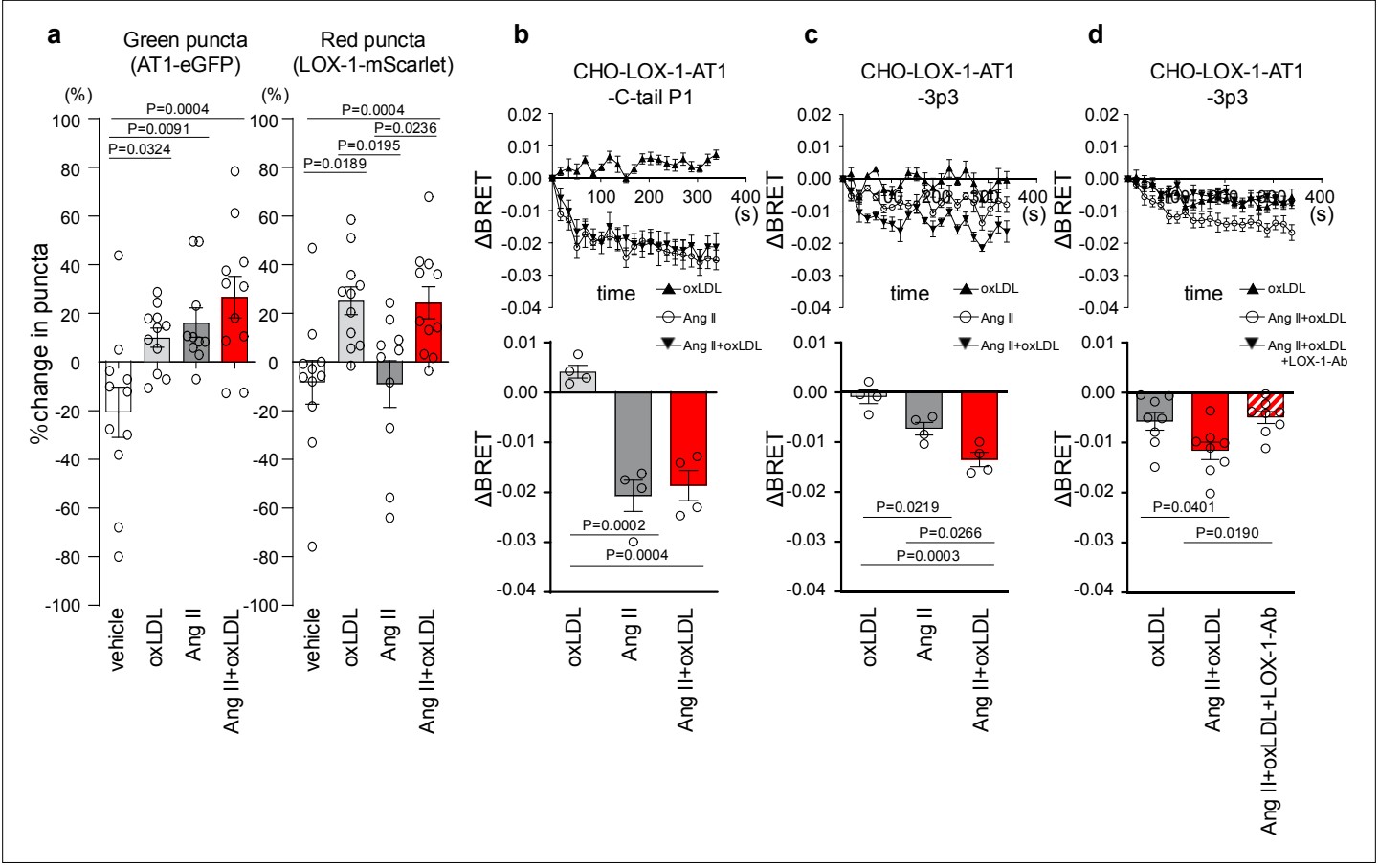

**Figure 2.** Co-treatment of oxidized low-density lipoprotein (oxLDL) with angiotensin II (Ang II) induces conformational change of AT1 different from each treatment alone. (**a**) Change in green puncta (AT1-eGFP) and red puncta (LOX-1-mScarlet) by the treatment with vehicle, oxLDL (10 µg/mL), Ang II ($10^{-5}$M), and the combination of oxLDL and Ang II in Chinese hamster ovary (CHO) cell overexpressing these fluorescent protein-conjugated receptors (n=10–11 for each group). The puncta were manually counted by a blinded observer and the number of puncta at 0 and 3 min was determined (N0 and N3, respectively). The change in puncta was calculated as (N0-N3)/N0. (**b-d**) Changes in BRET signals were monitored in CHO-LOX-1 cells expressing the following conformational biosensors: AT1-C-tailP1 (**b**) and AT1-ICL3P3 (**c, d**) bearing FlAsH insertion at the cytoplasmic-terminal tail (C-tailP1) and the third intracellular loop (ICL3P3) of AT1, respectively, that interact with Renilla luciferase at the end of the cytoplasmic tail. Cells were subjected to treatments with vehicle, oxLDL (10 µg/mL), Ang II ($10^{-5}$M), the combination of Ang II and oxLDL, and the combination of AngII, oxLDL, and LOX-1 antibody. The BRET ratios were calculated every 16 s for a total of 320 s and the relative change in intramolecular BRET ratio (ΔBRET) was calculated by subtracting the average BRET ratio measured for cells stimulated with vehicle at each time point. Lower panels indicate average ΔBRET of all the time points during measurement. Data are represented as mean ± SEM. Differences were determined by one-way ANOVA, followed by Tukey's multiple comparison test (**a-d**).

The online version of this article includes the following source data and figure supplement(s) for figure 2:

**Source data 1.** Co-treatment of oxidized low-density lipoprotein (oxLDL) with angiotensin II (Ang II) induces conformational change of AT1 different from each treatment alone.

**Figure supplement 1.** Live-imaging analysis of membrane LOX-1 and AT1 in response to the co-treatment of oxidized low-density lipoprotein (oxLDL) with AngII.

the concomitant binding of oxLDL to LOX-1 and Ang II to AT1 induced a conformational change in AT1 that was distinct from that induced by Ang II or oxLDL alone.

## Oxidized LDL potentiates Ang II-induced Gq-calcium signaling in renal cells

To confirm the pathophysiological significance of this phenomenon observed in the overexpressing cells, we validated it in cells endogenously expressing LOX-1 and AT1. We found that oxLDL in combination with Ang II did not increase the cellular IP1 content in human umbilical vein endothelial cells

(HUVECs), bovine vascular endothelial cells (BACEs), human aortic vascular smooth muscle cells (HAVSMCs), or rat macrophages (A10) (*Figure 3—figure supplement 1*). In contrast, in normal rat kidney epithelial cells (NRK52E) and fibroblasts (NRK49F), the combination of oxLDL and Ang II, but not Ang II alone, increased IP1 accumulation, which was suppressed in the presence of YM-25480, the Gq inhibitor (*Figure 3a and b*). IP1 accumulation induced by co-treatment with Ang II and oxLDL was abolished by the siRNA-mediated knockdown of either AT1 or LOX-1 (*Figure 3c and d*; the knock-down efficacy is shown in *Figure 3—figure supplement 2*). Regarding calcium influx, intracellular calcium concentrations were not increased by the treatment of either $10^{-7}$ M Ang II or low concentration of oxLDL (2 μg/ml) alone (*Figure 3e*, left and middle, *Figure 3f*). In contrast, the combination of Ang II ($10^{-7}$ M) and oxLDL (2 μg/ml) increased intracellular $Ca^{2+}$ concentration (*Figure 3e*, right, *Figure 3f*). The combined effect on calcium influx was attenuated by the siRNA knockdown of either AT1 or LOX-1 (*Figure 3g*). The Gq inhibitor and angiotensin receptor blocker (ARB), Irbesartan, also inhibited this phenomenon (*Figure 3h*). Calcium influx was not induced by either combination therapy or monotherapy in NRK52E cells (*Figure 3—figure supplement 3*).

## Co-treatment of oxLDL and Ang II enhanced cellular response upon Gq activation in renal cells

In NRK49F cells, co-treatment of oxLDL and Ang II, compared to vehicle, increased mRNA levels of NADPH components, *Ncf2* and *Cybb*, fibrosis markers, *Fn1*, *Col1a1*, *Col4a1*, and *Tgfb2*, and inflammatory cytokines, *Tnf*, *Il1b*, *Il6*, and *Ccl2* (*Figure 4a*). Notably, oxLDL did not alter the expression of the genes of interest, and Ang II increased the mRNA levels of only *Fn1* and *Ccl2*, indicating the apparent synergistic effect of co-treatment with Ang II and oxLDL on specific gene expression. This amplification effect of co-treatment was also observed in limited genes including *Ncf2*, *Tgfb2*, *Il6*, and *Ccl2* in NRK52E cells (*Figure 4b*). The combined effects of oxLDL and Ang II on gene expression were completely abolished by the Gq inhibitor (*Figure 4c and d*) and ARB (*Figure 4e and f*) in NRK49F and NRK 52E. We also verified protein expression of αSMA as a molecular marker of epithelial-mesenchymal transition (EMT) upon the indicated stimulation for 3 d in rat kidney cells. The combination of Ang II ($10^{-7}$ M) and oxLDL (5 μg/mL) induced αSMA expression in NRK49F or NRK52E to the same extent or less than TGFβ, a major inducer of EMT, respectively (*Figure 5a and b*). The results of the combination treatment were strikingly different from that of oxLDL or Ang II treatment alone, which did not affect αSMA expression in both types of cells (*Figure 5c and d*). The induction of αSMA by the combined treatment was suppressed in the presence of a Gq inhibitor or ARB in both types of cells, suggesting the AT1-Gq-dependent pathway in this phenomenon (*Figure 5e, f, g and h*). As a final in vitro assay, the proliferative activity of NRK49F cells after 24 hr of treatment with Ang II and oxLDL, either alone or in combination, was measured using the BrdU assay (*Figure 6a*). When administered alone, Ang II and oxLDL increased and reduced BrdU incorporation, respectively. The reducing effect of oxLDL on proliferation was blocked by Irbesartan, but not by a Gq inhibitor, consistent with our findings that oxLDL induces biased activation of AT1, which favors Gi but not Gq signaling (*Takahashi et al., 2021*). Oxidized LDL potentiated the Ang II-induced increase in BrdU incorporation, which was blocked in the presence of a Gq inhibitor or an ARB (*Figure 6a*). The siRNA-mediated knockdown of AT1 completely abolished the effects of Ang II and oxLDL, either alone or in combination. Knockdown of LOX-1 did not affect the pro-proliferative effect of Ang II itself but blocked the enhanced proliferation induced by co-treatment with oxLDL (*Figure 6b*).

## Oxidized LDL-inducible diet exacerbates Ang II-induced renal dysfunction in wild-type mice, but not in LOX-1 knockout mice

To examine the relevance of this phenomenon in renal injury, we replicated the in vivo conditions that occurred during simultaneous stimulation with oxLDL and Ang II in wild-type (WT) and LOX-1 knockout (LOX-1 KO) mice. This was achieved by inducing oxLDL via a high-fat diet (HFD) (*Kobori et al., 2009*) and Ang II via a subcutaneously implanted osmotic mini-pump (*Figure 7a*). Two different doses of Ang II were used in the experiment: a pressor dose of 0.7 γ, which was demonstrated to increase blood pressure (BP) and induce renal dysfunction (*Jennings et al., 2012*; *Wolak et al., 2009*), and a subpressor dose of 0.1 γ, which does not affect BP. For the intended analysis, the results are presented separately for mice exposed to subpressor and pressor doses of Ang II, utilizing the same outcomes of control animals infused with saline for comparison. Consistent with a previous report (*Fujita et al.,*

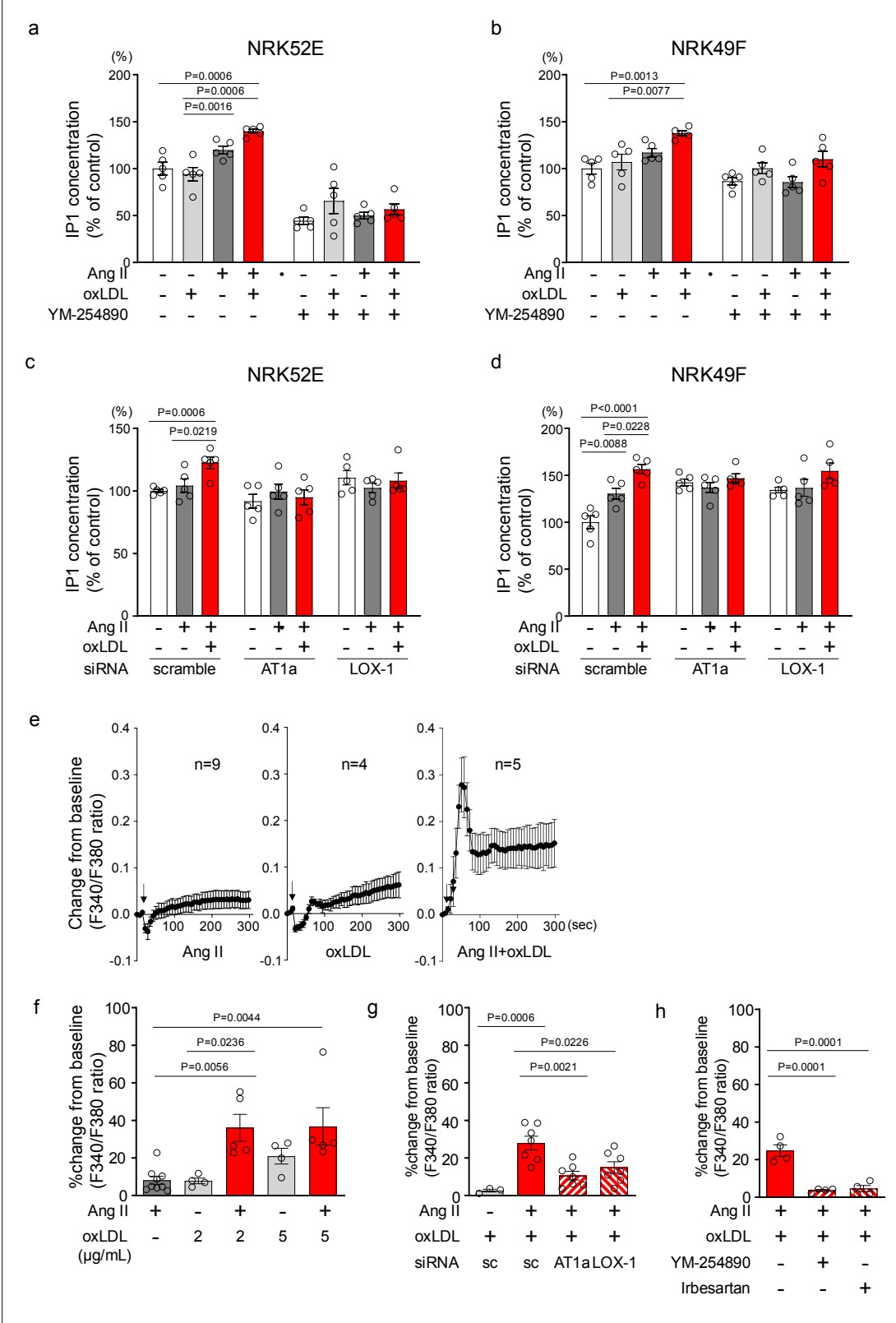

**Figure 3.** Oxidized low-density lipoprotein (LDL) potentiates angiotensin II (Ang II)-induced G protein αq subunit (Gq)-calcium signaling in renal cells. (**a, b**) IP1 concentration in response to Ang II ($10^{-7}$ M), oxLDL (10 μg/mL), and the combination of both with or without YM-254890, Gq inhibitor, in NRK52E (**a**) and NRK49F cells (**b**) (n=5 for each group). (**c, d**) IP1 concentration in response to Ang II ($10^{-7}$ M) and oxLDL (10 μg/mL) in the combination of Ang II ($10^{-7}$ M) and the additional effect of siRNA-mediated knockdown of *Agtr1a* or *Olr1* in NRK52E (**c**) and NRK49F cells (**d**) (n=5 for each group). (**e**)

*Figure 3 continued on next page*

*Figure 3 continued*

Intracellular calcium concentration in NRK49F cells using Fura 2-AM and dual-excitation microfluorometry. Changes in the fluorescence intensity ratio (F340/F380) served as an index of the calcium dynamics. Cells were exposed to Ang II ($10^{-7}$ M), oxLDL (2 µg/mL), and a combination of both agents. Addition of these agonists is marked with arrows on the timeline of the assay. Data acquisition and analysis were performed using a digital image analyzer to monitor real-time calcium flux (n=4–9). (**f**) Percentage changes from baseline in the ratio of emission signals (F340/F380) measured by Fura 2-AM were quantified following treatment with Ang II ($10^{-7}$ M) and oxLDL at the concentrations detailed in the Figure in NRK49F cells (n=5 for each group). Biological replicates were performed using 3 independent cell cultures. (**g**) Impact of siRNA-mediated knockdown *Agtr1a* or *Olr1* on the percentage changes from baseline in the ratio of emission signals (F340/F380) measured by Fura 2-AM were quantified following treatment with Ang II ($10^{-7}$ M) and oxLDL (2 µg/mL) in NRK49F cells (n=3–7). (**g**) Impact of co-treatment of YM-254890, Gq inhibitor, or Irbesartan, an angiotensin receptor blocker (ARB), on the percentage changes from baseline in the ratio of emission signals (F340/F380) measured by Fura 2-AM were quantified following treatment with Ang II ($10^{-7}$ M) and oxLDL (2 µg/mL) in NRK49F cells (n=4 for each group). Data are represented as mean ± SEM. Differences were determined using one-way ANOVA, followed by Tukey's multiple comparison test for (**a-d**) and (**f-h**).

The online version of this article includes the following source data and figure supplement(s) for figure 3:

**Source data 1.** Oxidized low-density lipoprotein (LDL) potentiates angiotensin II (Ang II)-induced G protein αq subunit (Gq)-calcium signaling in renal cells.

**Figure supplement 1.** Oxidized low-density lipoprotein (LDL) in combination with angiotensin II (Ang II) do not increase cellular IP1 content in human umbilical vein endothelial cells and bovine vascular endothelial cells, human aortic vascular smooth muscle cells, and rat macrophages.

**Figure supplement 1—source data 1.** Oxidized low-density lipoprotein (LDL) in combination with angiotensin II (Ang II) do not increase cellular IP1 content in human umbilical vein endothelial cells and bovine vascular endothelial cells, human aortic vascular smooth muscle cells, and rat macrophages.

**Figure supplement 2.** Efficiency of small interfering RNA (siRNA)-mediated knockdown for *AT1a* and *LOX-1* in NRK52E and NRK49F cells.

**Figure supplement 2—source data 1.** Efficiency of small interfering RNA (siRNA)-mediated knockdown for *AT1a* and *LOX-1* in NRK52E and NRK49F cells.

**Figure supplement 3.** Calcium influx was not induced by either the combination treatment of angiotensin II (Ang II) or oxidized low-density lipoprotein (oxLDL) or each treatment alone in NRK52E cells.

**Figure supplement 3—source data 1.** Calcium influx was not induced by either the combination treatment of angiotensin II (Ang II) or oxidized low-density lipoprotein (oxLDL) or each treatment alone in NRK52E cells.

*2009*), the HFD used in the study prominently increased the plasma LOX-1 ligand concentration, which was undetectable under a normal diet (ND) after 6 wk of feeding (*Figure 7—figure supplement 1*). Notably, the HFD did not intensify but rather attenuated the body weight increase during the experimental period compared to the ND (*Figure 7—figure supplement 2a–c*), likely due to reduced food intake in mice fed the HFD used in this study (*Figure 7—figure supplement 2d, e*). The pressor dose of Ang II similarly increased BP in HFD-fed and ND-fed WT mice (*Figure 7b*, *Figure 7—figure supplement 2f*). However, notably, there was a modest trend of attenuated BP elevation by 0.7 γ Ang II infusion in LOX-1 KO mice, in line with the previous report showing reduced Ang II-induced BP elevation by LOX-1 deficiency in mice (*Figure 7b*, *Figure 7—figure supplement 2f*; *Hu et al., 2008*). As expected, a subpressor dose of Ang II did not alter BP in the corresponding mouse group (*Figure 7b*, *Figure 7—figure supplement 2g*). Regarding biofluid analysis, HFD with saline infusion did not alter the urinary 8-OHdG concentration as a marker of oxidative stress and urinary albumin excretion (UAE) compared to ND with saline infusion in either WT or LOX-1 KO mice (*Figure 7c and d*). The pressor dose of Ang II with ND significantly increased the urinary 8-OHdG concentration and UAE in WT mice (*Figure 7c and d*). Notably, HFD feeding in WT mice with a pressor dose of Ang II resulted in a prominent increase in urinary 8-OHdG concentration and UAE compared to mice fed with ND (*Figure 7c and d*). In WT mice administered with a subpressor dose of Ang II, a significant increase in UAE was observed when comparing the effects of HFD to ND (*Figure 7d*). There were no significant differences in urinary 8-OHdG levels between the two dietary conditions (*Figure 7c*). Interestingly, when LOX-1 KO mice were fed either HFD or ND and then administered the corresponding dose of Ang II, no differences were observed in the measured parameters (*Figure 7c and d*). These findings indicate that the combination of HFD and Ang II administration appears to have a more pronounced effect on certain biofluid markers of renal injury in WT mice than in LOX-1 KO mice. The presence or absence of LOX-1 appears to influence the interaction between HFD and Ang II, affecting these specific parameters in mice. We did not find a difference in plasma aldosterone concentration between ND- and HFD-fed WT mice with a pressor dose of Ang II (*Figure 7—figure supplement 3*).

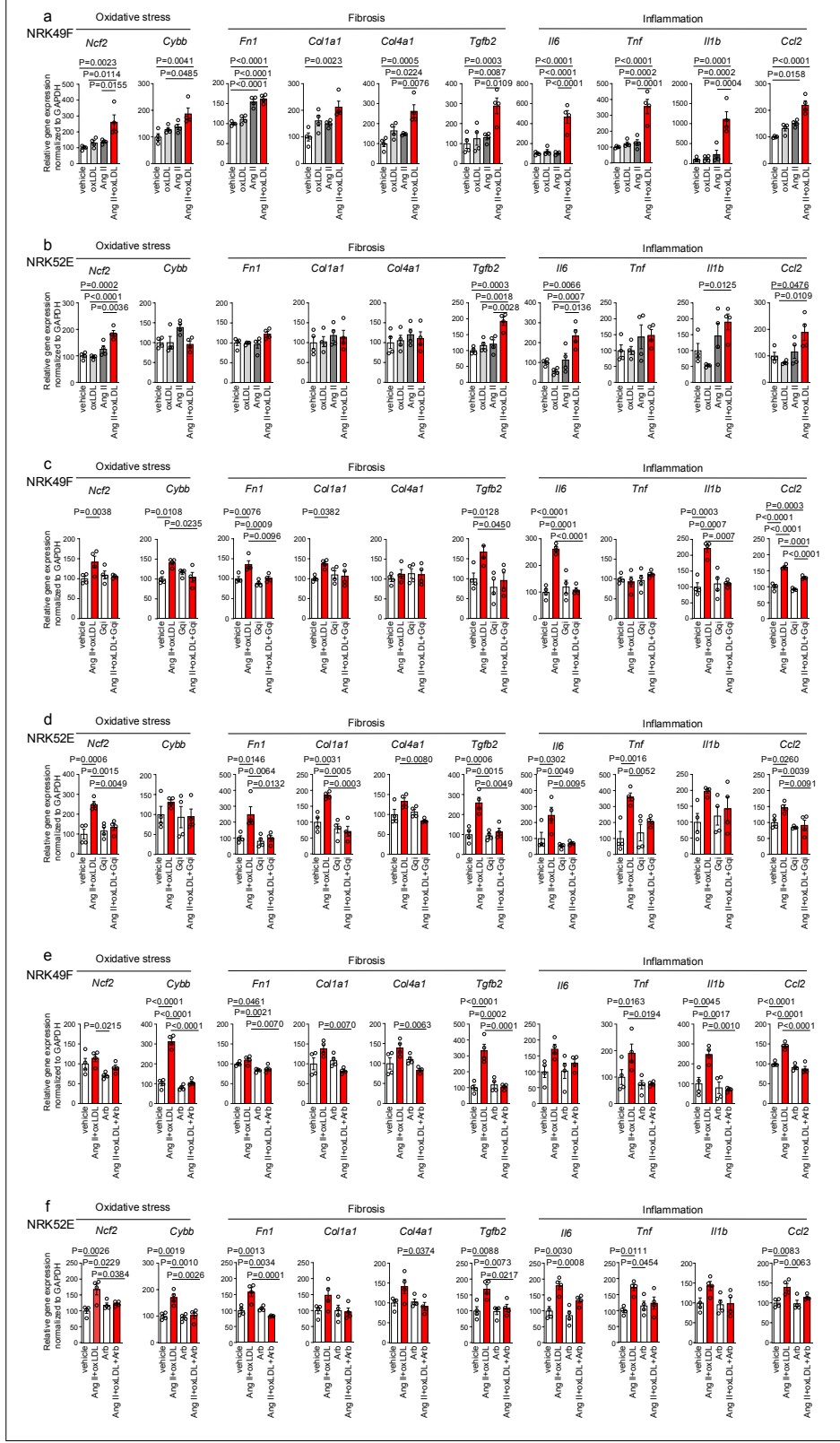

**Figure 4.** Co-treatment of oxidized low-density lipoprotein (oxLDL) and AII enhanced cellular response upon G protein αq subunit (Gq) activation in renal cells. (**a, b**) Quantitative real-time PCR analysis was performed to measure the gene expression of NADPH oxidase subunits (*Ncf2* and *Cybb*), fibrosis markers (*Fn1*, *Col1a1*, *Col4a1*, and *Tgfb2*), and inflammatory cytokines (*Tnf*, *IL1β*, *IL-6*, and *Ccl2*) in NRK49F cells (**a**) and NRK 52E cells (**b**). Gene

*Figure 4 continued on next page*

*Figure 4 continued*

expression levels were normalized to those of *Gapdh*. Cells were stimulated by oxLDL (5 μg/mL), angiotensin II (Ang II) ($10^{-7}$ M), or their combination (n=4 for each group). (**c, d**) Cells were pre-treated with vehicle or Gq inhibitor (YM-254890, Gqi), followed by treatment with vehicle or the combination of oxLDL (5 μg/mL) and Ang II ($10^{-7}$ M) in NRK49F cells (**c**) and NRK 52E cells (**d**) (n=4 for each group). (**e, f**) Cells were pre-treated with vehicle or ARB (Irbesartan, Arb), followed by treatment with vehicle or the combination of oxLDL (5 μg/mL) and Ang II ($10^{-7}$ M) in NRK49F cells (**e**) and NRK 52E cells (**f**) (n=4 for each group). Data are represented as mean ± SEM. Differences were determined by one-way ANOVA, followed by Tukey's multiple comparison test (**a-f**).

The online version of this article includes the following source data for figure 4:

**Source data 1.** Co-treatment of oxidized low-density lipoprotein (oxLDL) and AII enhanced cellular response upon G protein αq subunit (Gq) activation in renal cells.

We then conducted quantitative real-time PCR analysis on genes within kidney sections, encompassing NADPH components (*Ncf1, Ncf2, Ncf4, and Cybb*), inflammatory cytokines (*Il6, Tnf, Il1b, Ccl2, and Ptgs2*), fibrosis markers (*Tgfb2, Fn1, Col1a1, Col4a1, Acta2, and Vim*), epithelial markers (*Cdh1 and Cdh16*), and tubular marker (*Lcn2*) (*Figure 8*, *Figure 8—figure supplement 1*). In mice treated with a pressor dose of Ang II, 15 out of 18 genes examined showed enhanced alterations in gene expression, indicative of heightened NADPH oxidase components, inflammatory cytokines, fibrosis, decreased epithelial markers, and exacerbated tubular injury in HFD-fed WT mice compared to ND-fed mice (*Figure 8a*, *Figure 8—figure supplement 1a*). For many of these genes, a synergistic effect of the combination of a pressor dose of Ang II and HFD was not observed in LOX-1 KO mice (*Figure 8a*, *Figure 8—figure supplement 1a*). Seven genes displayed discernible differences between HFD- and ND-fed WT mice treated with a subpressor dose of Ang II (*Figure 8b*, *Figure 8—figure supplement 1b*). Other than *fibronectin*, no differences were observed between ND- and HFD-fed LOX-1 mice exposed to a subpressor dose of Ang II (*Figure 8b*, *Figure 8—figure supplement 1b*). In relation to *Agtr1a* and *Olr1* expression, no variations emerged between the ND and HFD groups when subjected to the corresponding dose of Ang II treatment in both WT and LOX-1 KO mice (*Figure 8—figure supplement 1a, b*).

In the histological analysis of kidney samples, in contrast to the gene expression data, the administration of either a subpressor or pressor dose of Ang II, both in isolation and in combination with HFD for 4 wk, did not reveal any notable increase in fibrosis, as evaluated by Masson-Trichrome staining (*Figure 9—figure supplement 1a*). Similarly, there was no significant change in the degree of mesangial expansion or glomerular area, as assessed by PAS staining. (*Figure 9—figure supplement 1b*).

Finally, immunofluorescence staining of mouse kidney specimens was performed to detect the colocalization sites of LOX-1 and AT1 in the kidney. As shown in *Figure 9a and b*, LOX-1 and AT1a were predominantly colocalized in the renal tubules, but not in the glomeruli. We also performed co-immunofluorescence staining with megalin, a well-established marker of proximal renal tubules, as shown in *Figure 9—figure supplement 2*. Both AT1 and LOX-1 were observed to colocalize with megalin, especially at the brush borders, indicating their presence within the same renal compartments involved in AT1/LOX-1 signaling.

## Discussion

We conducted a series of experiments to provide empirical support for our hypotheses. We propose that the simultaneous binding of Ang II to AT1 and oxLDL to LOX-1 triggers distinct and more pronounced structural modifications in AT1 than the individual modifications induced by each ligand. This structural alteration in AT1 leads to the enhanced activation of Gq signaling.

To confirm AT1 and LOX-1 colocalization and interaction at the cell membrane, our previous study used in situ proximity ligation assays (PLA) and membrane protein immunoprecipitation assays in CHO cells expressing tagged AT1 and LOX-1, successfully demonstrating AT1/LOX-1 complex formation (*Yamamoto et al., 2015*). Additionally, we provided histological evidence of their co-localization with megalin in proximal renal tubules to support these findings, although the limitation remains regarding the lack of direct in vivo evidence for membrane co-localization of LOX-1 and AT1. While additional co-staining with other markers to identify specific cell types was not conducted, the prominent

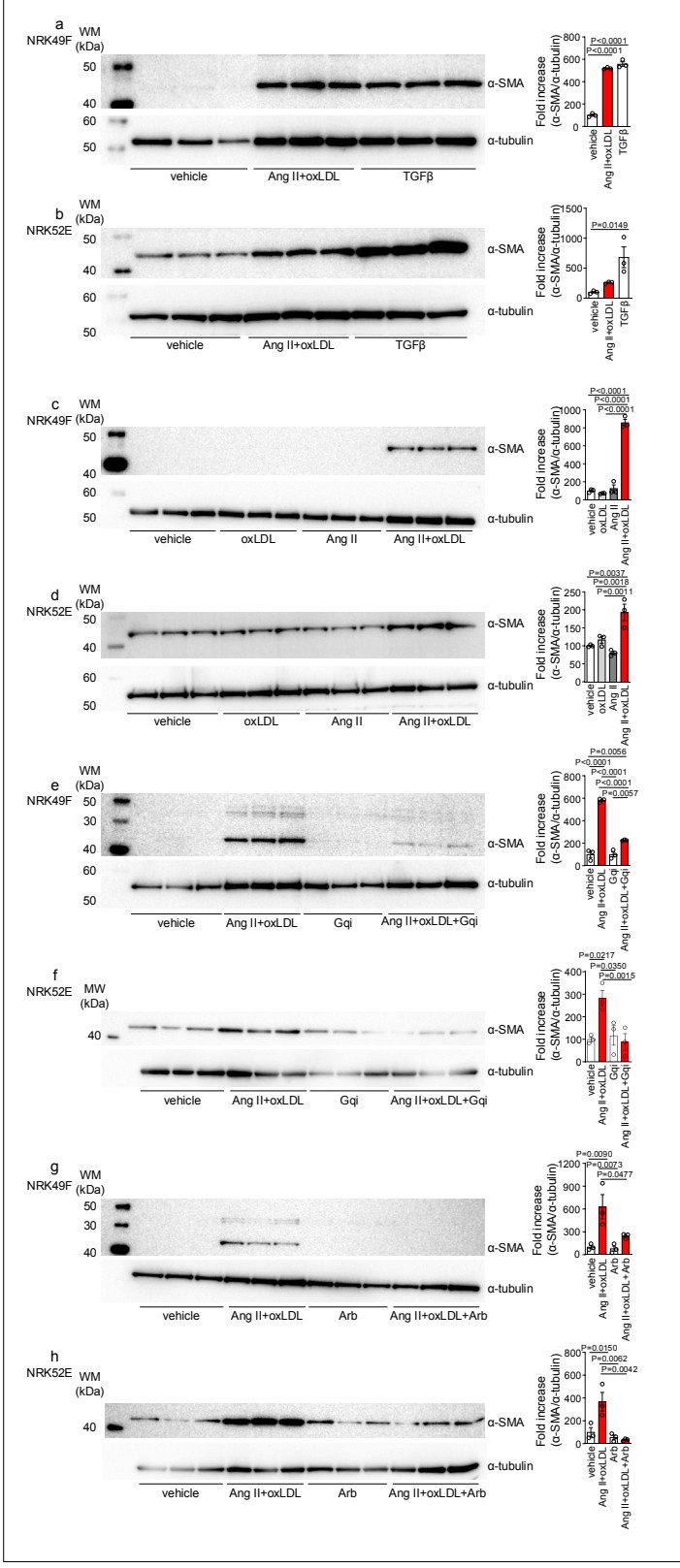

**Figure 5.** Oxidized low-density lipoprotein (LDL) enhanced angiotensin II (Ang II)-induced epithelial-mesenchymal transition (EMT) in NRK52E and NRK49F cells. (**a, b**) Left: Western blot analysis of α-smooth muscle actin (α-SMA), a marker of epithelial-mesenchymal transition (EMT), in NRK49F (**a**) and NRK52E (**b**) cells. Cells were stimulated with oxLDL (5 µg/mL), Ang II (10⁻⁷ M), and TGF-β (10 ng/mL), with TGF-β serving as a well-known EMT inducer.

*Figure 5 continued on next page*

*Figure 5 continued*

Right: Densitometric analysis of α-SMA protein expression normalized to α-Tubulin (n=3 for each group). (**c, d**) Left: Western blot analysis of α-SMA in NRK49F (**c**) or NRK52E (**d**) after stimulation with oxLDL (5 µg/mL), Ang II (10$^{-7}$ M), and their combination. Right: Densitometric analysis of α-SMA protein expression normalized to α-tubulin (n=3 for each group). (**e, f**) Left: Western blot analysis of α-SMA in NRK49F (**e**) or NRK52E (**f**) after treatment with a combination of oxLDL (5 µg/mL) and Ang II (10$^{-7}$ M). Prior to this treatment, the cells were pre-treated with either a vehicle or a Gq inhibitor (YM-254890, Gqi). Right: Densitometric analysis of α-SMA protein expression normalized to α-Tubulin (n=3 for each group). (**g, h**) Left: Western blot analysis of α-SMA in NRK49F (**e**) or NRK52E (**f**) after treatment with a combination of oxLDL (5 µg/mL) and Ang II (10$^{-7}$ M). Prior to treatment, cells were pre-treated with either vehicle or ARB (Irbesartan, Arb). Right: Densitometric analysis of α-SMA protein expression normalized to α-tubulin (n=3 for each group). Data are represented as mean ± SEM. Differences were determined by one-way ANOVA, followed by Tukey's multiple comparison test (**a-f**).

The online version of this article includes the following source data for figure 5:

**Source data 1.** Oxidized low-density lipoprotein (LDL) enhanced angiotensin II (Ang II)-induced epithelial-mesenchymal transition in NRK52E and NRK49F cells.

**Source data 2.** Oxidized low-density lipoprotein (LDL) enhanced angiotensin II (Ang II)-induced epithelial-mesenchymal transition in NRK52E and NRK49F cells.

**Source data 3.** Oxidized low-density lipoprotein (LDL) enhanced angiotensin II (Ang II)-induced epithelial-mesenchymal transition in NRK52E and NRK49F cells.

localization of AT1R with megalin in our study provides strong evidence of its expression in proximal renal tubules, consistent with established findings regarding AT1R presence in this nephron segment. Previous studies have documented AT1R expression in various renal cells, including mesangial, interstitial, and juxtaglomerular (JG) cells, as well as proximal tubules. In our immunofluorescence analysis, however, we did not observe significant AT1R expression in the glomerulus or mesangium. The pronounced expression of AT1R in proximal tubules aligns with previous reports (*Arthur et al., 2021*), though limitations in immunofluorescence sensitivity do not exclude AT1R presence in other compartments. Notably, our focus on proximal tubules enabled clear observation of AT1/LOX-1 co-localization, especially under oxLDL and AngII stimulation. This interaction underscores a potential focal point for AT1R/LOX-1 signaling in kidney disease pathogenesis within the renal system.

At the cellular level, live imaging analysis (*Figure 2—figure supplement 1*) displayed a limited number of overlapping LOX-1 and AT1R puncta, which could be attributed to the dynamic and transient nature of the LOX-1 and AT1 receptor interaction. As described in our previous study (*Yamamoto et al., 2015*), this interaction is sensitive to buffer conditions, with complex formation occurring under non-reducing but not reducing conditions. This indicates that the LOX-1 and AT1 interaction is mediated by non-covalent rather than stable covalent bonds, leading these receptors to form and dissociate rapidly. Consequently, only a small fraction of LOX-1 and AT1 receptors may be co-localized at any given time point, explaining the limited number of overlapping puncta observed in live imaging. Importantly, despite this limited spatial overlap, our findings indicate that co-treatment with oxLDL and Ang II significantly enhances Gq signaling. This highlights that the functional impact of the LOX-1/AT1 interaction, especially in response to specific stimuli like oxLDL and Ang II, is more crucial to downstream signaling outcomes than the extent of stable receptor co-localization.

The current experiments showed that oxLDL enhanced the effects of Ang II-induced production of IP1, a downstream signaling molecule associated with Gq activation and subsequent calcium influx, further supporting functional activation of the AT1/LOX-1 complex. However, this effect was observed only in the presence of both LOX-1 and AT1. The differences in activation levels observed between the IP1 assay and the calcium influx assay, both indicators of Gq activity, likely arise due to variations in assay sensitivity. While the absolute differences in the IP1 assay between treatment groups may appear modest, the critical comparison between Ang II alone and Ang II with oxLDL consistently demonstrated significant differences, in alignment with the calcium influx results. Notably, co-treatment with oxLDL reduced the EC50 for IP1 production by 80% compared to Ang II alone, underscoring a robust enhancement of Gq signaling, even though the IP1 assay differences were relatively small in absolute terms.

Considering that oxLDL selectively activates Gi but not Gq through the LOX-1-AT1 dependent pathway (*Takahashi et al., 2021*), it is evident that the observed phenomenon cannot be attributed

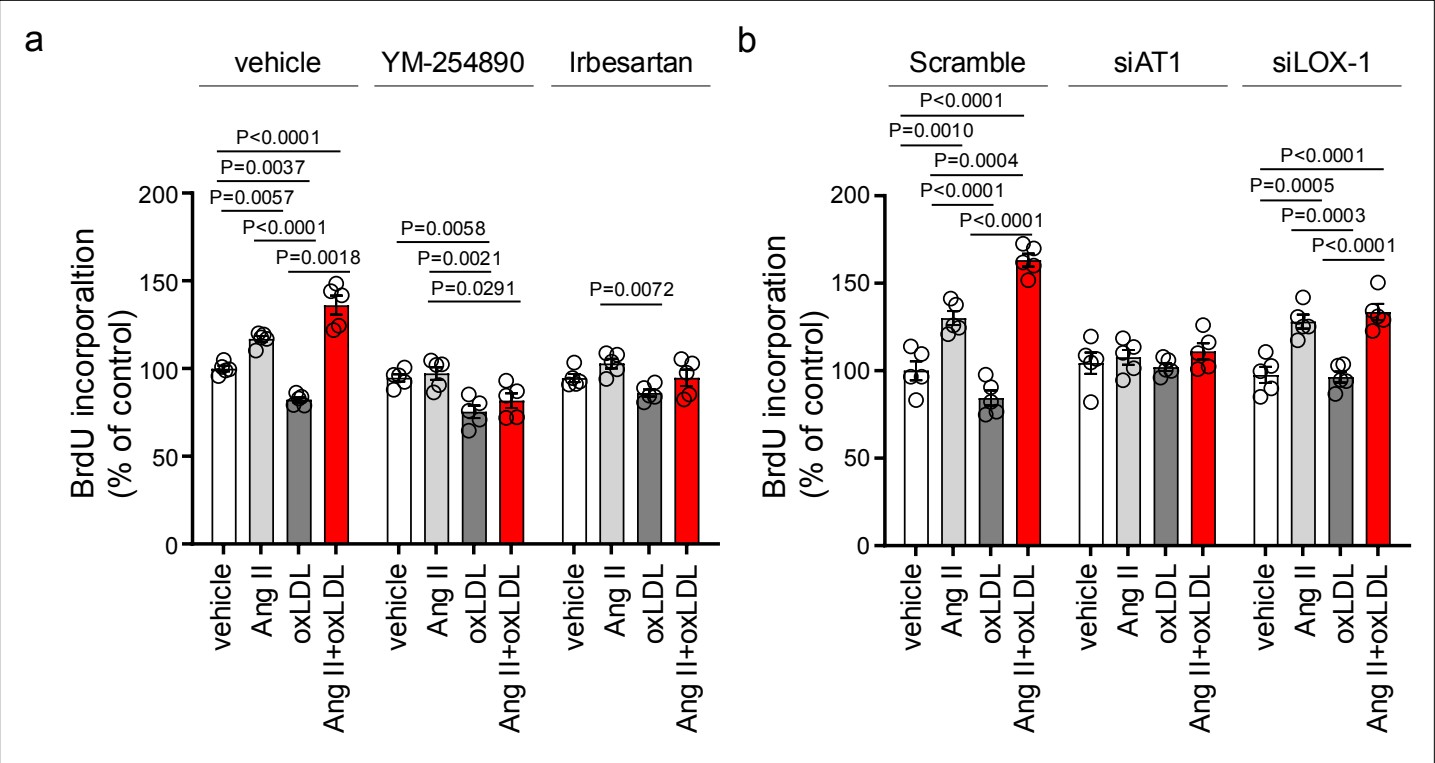

**Figure 6.** Oxidized low-density lipoprotein (LDL) enhanced angiotensin II (Ang II)-induced renal fibroblast proliferation via AT1-G protein αq subunit (Gq) signaling and LOX-1-dependent manner. (**a**) Proliferative activity assessed by BrdU incorporation into NRK49F cells. Cells were pretreated with vehicle, YM-254890, or ARB, Irbesartan, followed by the treatment with oxLDL (5 μg/mL), Ang II ($10^{-7}$ M), or their combination. (n=5 for each group). Biological replicates were performed using two independent cell cultures. (**b**) NRK49F cells were subjected to siRNA-mediated knockdown using specific siRNAs for *Agtr1a* (siAT1) or *Olr1* (siLOX-1). Following knockdown, cells were treated with either vehicle, oxLDL (5 μg/mL), Ang II ($10^{-7}$ M), or their combination. Proliferative activity was assessed by measuring the BrdU levels (n=5 for each group). Data are represented as mean ± SEM. Differences were determined using one-way ANOVA, followed by Tukey's multiple comparison test for (**a**) and (**b**). Biological replicates were performed using two independent cell cultures.

The online version of this article includes the following source data for figure 6:

**Source data 1.** Oxidized low-density lipoprotein (LDL) enhanced angiotensin II (Ang II)-induced renal fibroblast proliferation via AT1-G protein αq subunit (Gq) signaling and LOX-1-dependent manner.

solely to the additive effect of oxLDL on AT1 activation. Rather, the simultaneous binding of oxLDL and Ang II to their respective receptors, LOX-1 and AT1, which form a single complex, underlies this phenomenon. Indeed, our findings from live imaging of the membrane receptors revealed that the combination of Ang II and oxLDL did not induce any additional influence on the internalization of both receptors upon AT1-β-arrestin activation compared to each ligand alone, suggesting that the quantity of activated receptors is not affected by the combination treatments. Considering the conformation-activation relationship in AT1 activation (*Devost et al., 2017*), this supports the hypothesis that the simultaneous binding of Ang II and oxLDL in a single AT1-LOX-1 complex induces a greater conformational change, resulting in a more open conformation of AT1 compared with each ligand alone. However, it is technically challenging to directly identify structural modifications of the receptor complex using techniques such as crystal structure analysis. Instead, we utilized AT1 conformational sensors capable of differentiating between the conformational changes induced by Ang II and biased AT1 (*Devost et al., 2017*). Interestingly, we observed that one of the two conformational sensors employed in this study detected an augmented response in the presence of the combination treatment compared with Ang II alone. Importantly, this enhanced response was significantly inhibited when an LOX-1 antibody was introduced, indicating the dependency of LOX-1 on this phenomenon. These results indicate that the combined treatment with Ang II and oxLDL in the presence of LOX-1 induces a unique conformational change in individual AT1 molecules, which differs from the conformational changes induced by each single treatment alone.

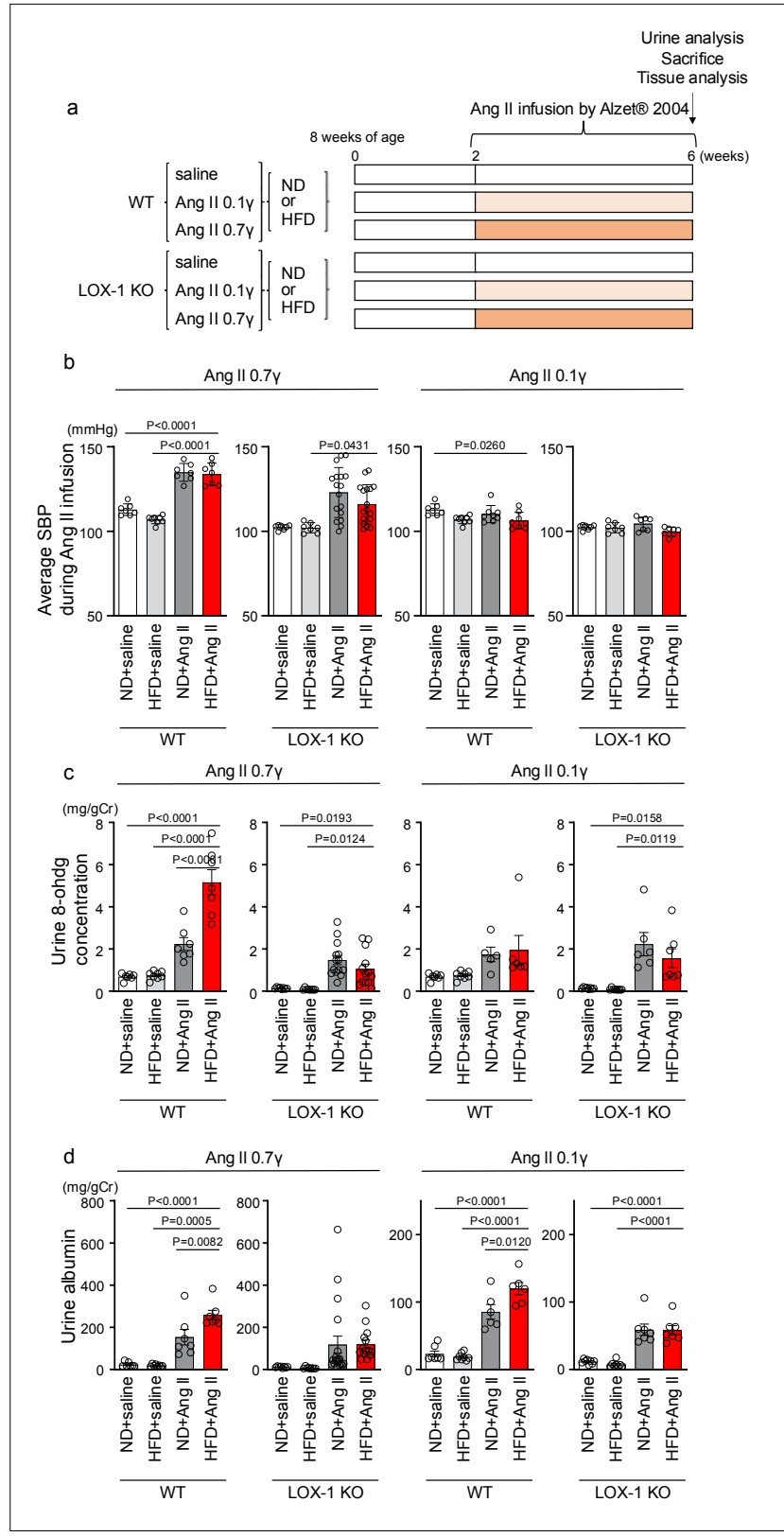

**Figure 7.** Oxidized low-density lipoprotein (LDL)-inducible diet exacerbates angiotensin II (Ang II)-induced renal dysfunction in wild-type mice, but not in LOX-1 knockout mice. (**a**) A Schematic protocol for the animal experiments. Eight-wk-old male wild-type (WT) mice and male LOX-1 KO mice were fed either an normal diet (ND) or an high-fat diet (HFD) for 6 wk. After 10 wk of age, the mice were treated over a 4 wk period with infusions

*Figure 7 continued on next page*

*Figure 7 continued*

of either saline or Ang II. Ang II was administered at two dosage levels: a subpressor dose of 0.1 γ and a pressor dose of 0.7 γ, delivered via subcutaneously implanted osmotic pumps. At the end of the infusion period, urine was collected, the animals were sacrificed, and comprehensive tissue analysis was conducted to evaluate the renal effects of the treatments. (**b**) Average systolic blood pressure (SBP) measured at half-week intervals in WT and LOX-1 KO mice during the 4 wk infusion period. (**c, d**) Urine 8-OHDG concentrations (mg/g creatinine [Cr]) (**c**) and urine albumin concentrations (mg/g creatinine [Cr]) (**d**) in WT and LOX-1 KO mice at the conclusion of the 4 wk infusion period. Data are represented as mean ± SEM. Differences were determined by one-way ANOVA, followed by Tukey's multiple comparison test (**a-d**).

The online version of this article includes the following source data and figure supplement(s) for figure 7:

**Source data 1.** Oxidized low-density lipoprotein (LDL) inducible diet exacerbates angiotensin II (Ang II)-induced renal dysfuntion in wild-type mice, but not in LOX-1 knockout mice.

**Figure supplement 1.** High-fat Diet used in the study prominently increased plasma LOX-1 ligand concentration.

**Figure supplement 1—source data 1.** High-fat diet used in the study prominently increased plasma LOX-1 ligand concentration.

**Figure supplement 2.** Impact of Diet and angiotensin II (Ang II) Infusion on Body Weight and Systolic Blood Pressure in Mice.

**Figure supplement 2—source data 1.** Impact of diet and angiotensin II (Ang II) infusion on body weight and systolic blood pressure in mice.

**Figure supplement 3.** No significant difference was found in plasma aldosterone concentration between a normal diet and a high fat diet-fed wild-type mice with a pressor dose of angiotensin II (Ang II).

**Figure supplement 3—source data 1.** No significant difference was found in plasma aldosterone concentration between a normal diet and a high fat diet-fed wild-type mice with a pressor dose of angiotensin II (Ang II).

It is important to note that the ability of LOX-1 ligands to enhance Ang II-AT1 signaling is not commonly observed. This was corroborated by the observation that BSA-conjugated AGE, a recognized ligand of LOX-1 (*Lymperopoulos et al., 2021*), failed to augment the production of IP1 by Ang II compared to control BSA (*Figure 1d*). While the exact reason for this discrepancy is not yet understood, it is noteworthy that the predicted particle size of oxLDL (*Ohki et al., 2005*) is significantly larger at 250 Å compared to the maximum particle size of AGE-BSA (*Wright and Thompson, 1975*), which is 120 Å. This suggests a potentially greater impact of oxLDL on the structural modifications within the AT1-LOX-1 complex. However, further structural analysis is required to validate this hypothesis.

Notably, the Gq bias resulting from combination treatment varied across the mammalian cells examined. Specifically, we observed a combinatorial effect exclusively in renal epithelial and fibroblasts, whereas vascular endothelial and smooth muscle cells did not display the same response. In these renal cells, we observed increased Gq signaling, along with other cellular phenomena such as calcium influx, changes in gene expression, and alterations in cellular characteristics, including myofibroblast activation and cell proliferation. While our study demonstrates a significant upregulation of α-SMA, a well-established marker of myofibroblast, in renal epithelial and fibroblast cells exposed to combined Ang II and oxLDL treatment, we acknowledge the evolving understanding of EMT, particularly the role of partial epithelial-mesenchymal transition (pEMT) in CKD. Notably, pEMT has garnered attention under conditions of inflammation, oxidative stress, and elevated TGF-β, which are also relevant to our Ang II and HFD models. Unlike full EMT, where epithelial cells completely transition into mesenchymal cells, pEMT represents a state in which epithelial cells partially acquire mesenchymal characteristics, such as increased α-SMA expression and the secretion of pro-fibrotic cytokines, while remaining attached to the basement membrane without fully transitioning into fibroblasts (*Sheng and Zhuang, 2020*). Importantly, previous studies using the unilateral ureteral obstruction (UUO) model suggest that full EMT is unlikely to play a significant role in renal fibrosis and that most kidney fibroblasts are thought to originate from interstitial cells rather than via EMT (*Kriz et al., 2011*). The observed increase in α-SMA in our model may, therefore, indicate a pEMT-like state, indirectly contributing to fibrosis by promoting cytokine and growth factor release rather than directly driving fibroblast generation. This interpretation aligns with findings from other kidney fibrosis models, including the UUO model, which shares pathophysiological features such as inflammation and oxidative stress with our model. Given these considerations, the increased α-SMA expression observed in our study

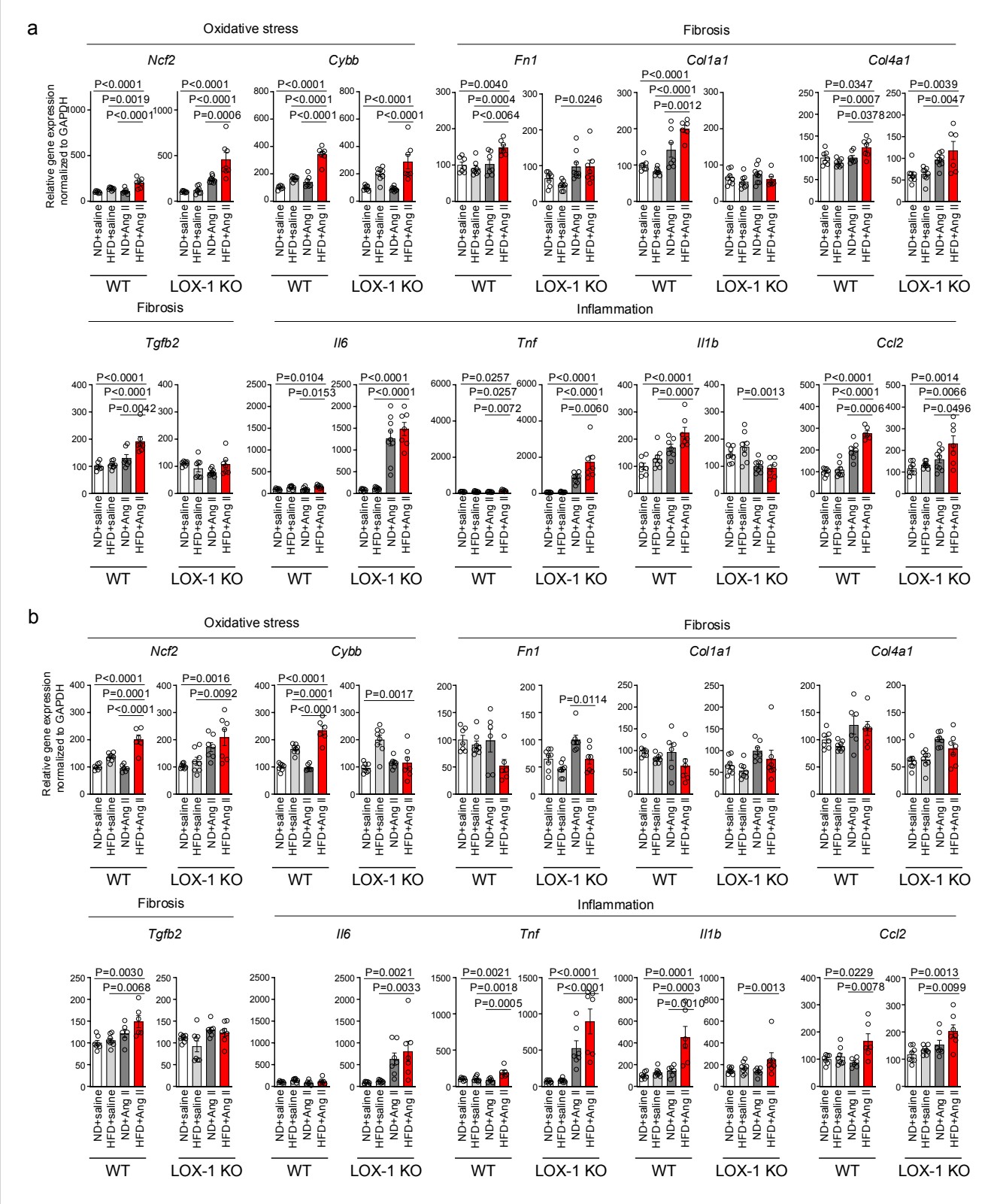

**Figure 8.** A high-fat diet enhanced angiotensin II (Ang II)-induced renal injury-related gene expression in the kidney in a LOX-1-dependent manner. Quantitative real-time PCR analysis for gene expression of NADPH components (*Ncf2 and Cybb*), inflammatory cytokines (*Il6, Tnf, Il1b, and Ccl2*), and fibrosis markers (*Tgfb2, Fn1, Col1a1, and Col4a1*) in the kidney harvested from wild-type (WT) and LOX-1 KO mice. The experimental procedures, including the dietary regimen and Ang II administration, are detailed in **Figure 7a**. Mice were administered either a pressor dose of 0.7 γ Ang II (**a**) or

*Figure 8 continued on next page*

*Figure 8 continued*

a subpressor dose of 0.1 γ Ang II (**b**). Data are represented as mean ± SEM. Differences were determined using one-way ANOVA, followed by Tukey's multiple comparison test for (**a**) and (**b**).

The online version of this article includes the following source data and figure supplement(s) for figure 8:

**Source data 1.** A high-fat diet enhanced angiotensin II (Ang II)-induced renal injury-related gene expression in the kidney in a LOX-1-dependent manner.

**Figure supplement 1.** A high-fat diet enhanced angiotensin II (Ang II)-induced renal injury-related gene expression in the kidney in a LOX-1-dependent manner.

**Figure supplement 1—source data 1.** A high-fat diet enhanced angiotensin II (Ang II)-induced renal injury-related gene expression in the kidney in a LOX-1-dependent manner.

may be indicative of pEMT rather than definitive evidence of EMT directly contributing to fibroblast differentiation. Additionally, extrapolating in vitro pEMT findings to in vivo models presents inherent challenges, as detecting these subtle phenotypic changes remains complex (*Hadpech and Thongboonkerd, 2024*), (*Sheng and Zhuang, 2020*). Further mechanistic investigations are required to clarify the contribution of pEMT to renal fibrosis. Nevertheless, our findings support the possibility that pEMT may contribute to fibrosis within the specific pathophysiological context of Ang II and HFD co-administration.

Additionally, fibroblast proliferation, as assessed by the BrdU assay, was notably enhanced by the combined treatment, as opposed to each treatment administered separately. Interestingly, the use of 5 µg/mL oxLDL in our study showed a tendency to decrease proliferation, which was counteracted by the administration of an ARB but not a Gq inhibitor. These findings suggest that the presence of Ang II leads to a significant transformation in the function of oxLDL, primarily due to its altered influence within the AT1-LOX-1 complex. Taken together, these results suggest that the simultaneous binding of oxLDL to LOX-1 and Ang II to AT1 results in a Gq-biased shift in AT1 activation, leading to a cellular phenomenon that could potentially contribute to renal fibrosis.

In an animal study, we introduced a biological environment in which both circulating Ang II and oxLDL (an LOX-1 ligand) were increased in mice. Previous studies using mice fed an HFD have consistently reported the onset of renal injury, as determined by various measurements (*Jiang et al., 2005*; *Kume et al., 2007*; *Yamamoto et al., 2017*; *Sun et al., 2020*; *Yu et al., 2022*). In contrast, 6 wk of HFD feeding without Ang II treatment did not alter renal function in our mice. This can be attributed to the lack of obesity induced by the HFD in this study. We used this diet based on a previous study that confirmed increased circulating LOX-1 ligand levels without body weight gain in mice (*Sato et al., 2008*). In our experiments, this diet led to a decrease in body weight, likely due to reduced food intake in HFD-fed mice (*Figure 7—figure supplement 1d, e*). Although modest, this weight reduction may influence renal function. Obesity is a well-established and clinically proven risk factor of renal dysfunction. The mechanisms underlying this association are complex and involve various factors other than lipid abnormalities, such as hemodynamic changes that affect kidney circulation and the impact of adipose tissue on the production of adipokines and other inflammatory mediators (*García-Carro et al., 2021*; *Tsuboi et al., 2017*). Consequently, we observed the influence of elevated lipid particle levels on renal function, independent of obesity. We found that the effect of an HFD became obvious with an increase in the Ang II load in WT mice. In particular, in WT mice treated with high-dose Ang II, which elevated systolic BP by approximately 30 mm Hg, an HFD induced notable increases in urinary reactive oxygen species and urinary albumin. In contrast, an HFD had no impact on Ang II-infused LOX-1 KO mice, as evidenced by equivalent urinalysis measurements for renal injury between the diets. Correspondingly, simultaneous administration of an HFD and Ang II resulted in a consistent alteration in the expression of genes related to renal injury, including fibrosis, inflammation, and oxidative stress, except for some genes in WT mice, but not in LOX-1 KO mice. This strongly suggests that the combined effect of Ang II and HFD on renal function is LOX-1 dependent. Nevertheless, it should be noted that the effect of high-dose Ang II infusion on BP tended to be less pronounced in LOX-1 KO mice compared to WT mice, although there were no differences in BP elevation between the diets in each group of mice. This reduction in BP may contribute to the decreased renal injury observed in LOX-1 KO mice, independent of the AT1/LOX-1 interaction. These findings align with those of previous studies indicating that LOX-1 knockout mice show resistance to Ang II-induced elevation of

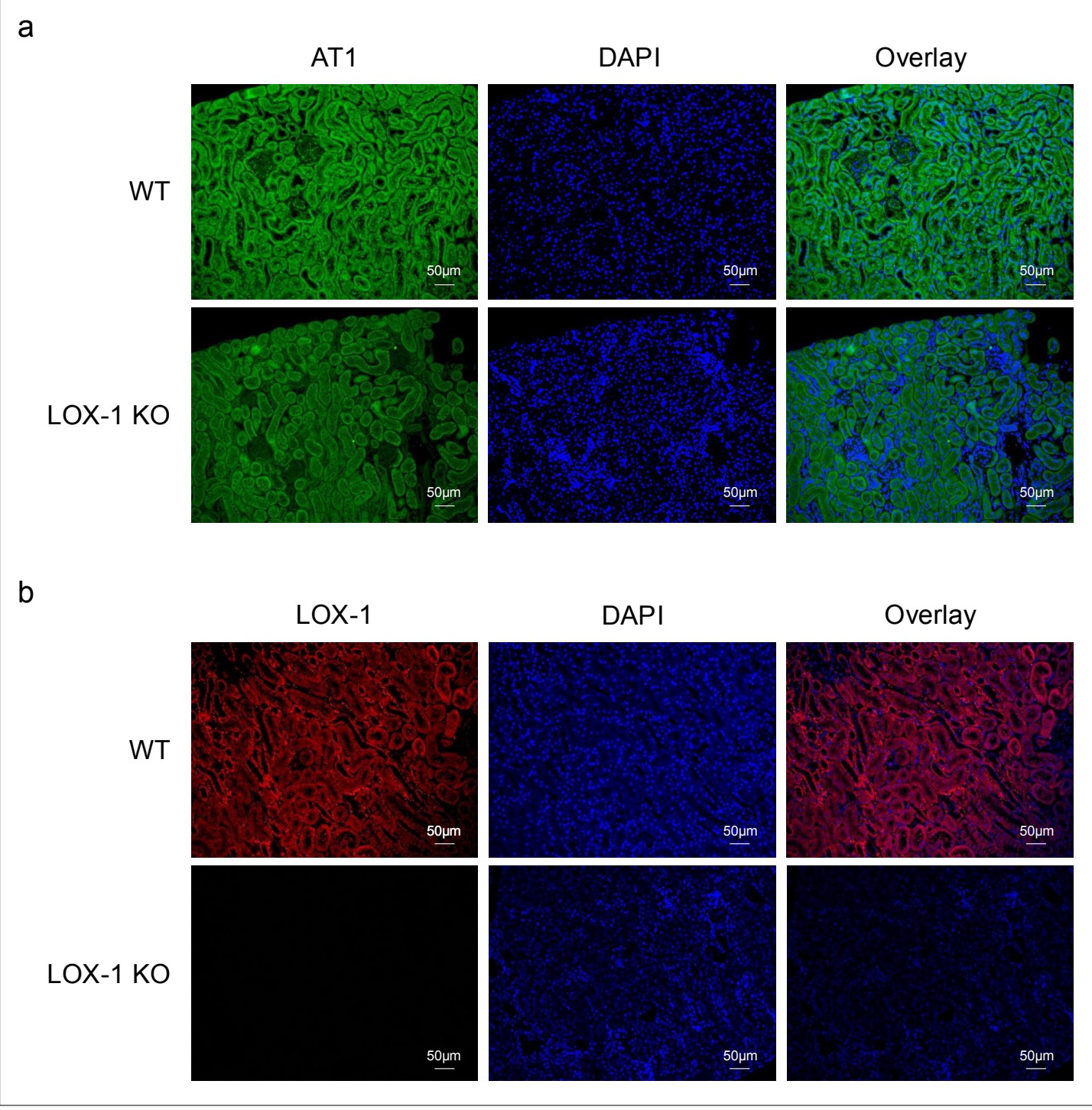

**Figure 9.** LOX-1 and AT1a were predominantly co-localized in renal tubules. (**a, b**) Representative images depicting staining for LOX-1 (a) and AT1 (**b**) in the renal cortex tissues from wild-type mice (WT) and LOX-1 knockout mice (LOX-1 KO). Nuclei are stained blue with DAPI. Green and red signals indicate AT1, while the red signal indicates LOX-1. Overlay images demonstrate the merged visualization of AT1 or LOX-1 with DAPI, highlighting the predominant colocalization of LOX-1 and AT1 in renal tubules as opposed to the glomerulus.

The online version of this article includes the following source data and figure supplement(s) for figure 9:

**Figure supplement 1.** The treatment with angiotensin II (Ang II), a high fat diet, or their combination for 4 wk did not induce any histological changes indicative of renal injury.

**Figure supplement 1—source data 1.** The treatment with angiotensin II (Ang II), a high fat diet, or their combination for 4 wk did not induce any

*Figure 9 continued on next page*

*Figure 9 continued*

histological changes indicative of renal injury.

**Figure supplement 2.** LOX-1 and AT1a were co-localized with megalin.

BP (*Hu et al., 2008*; *Hu et al., 2009*; *Li et al., 2021*.) Specifically, when mice were infused with 2 γ Ang II (equivalent to 25 g mice) for a duration of 28 d, wild-type mice experienced a BP increase exceeding 180 mmHg, while LOX-1 knockout mice demonstrated a reduction of approximately 40 mmHg in this elevation (*Hu et al., 2008*). Furthermore, the same research group reported that Ang II infusion led to less severe renal injury in LOX-1KO mice compared to WT mice (*Hu et al., 2009*). Importantly, these findings were observed in mice fed with an ND, suggesting that the protective effect of LOX-1 loss-of-function against Ang II-induced elevated BP occurs through the LOX-1-AT1 complex, independent of the presence of oxLDL. Additionally, under subpressor doses of Ang II, where no significant differences in BP were observed, HFD-fed WT mice still exhibited increased renal injury compared to ND-fed mice, an effect that was reduced in LOX-1 KO mice. These findings suggest that the protective effects of loss-of-function of LOX-1 are partly independent of BP changes, underscoring the role of the AT1/LOX-1 interaction in renal injury with Ang II and HFD co-treatment. We recognize that our single time-point analysis (1.5 mo post-treatment) also may limit these observations, as the effects of AT1/LOX-1 interaction on renal injury could vary with treatment duration. Taken together, the effects of LOX-1 on AT1 signaling are complex, involving both ligand-dependent and -independent mechanisms, and further investigation is required for a comprehensive understanding of the LOX-1-AT1 interaction.

Regarding the effects of AT1 and LOX-1 interaction on the renin-angiotensin system (RAS) and blood pressure, oxLDL binding to LOX-1 enhances AT1 receptor-mediated Gq signaling, thereby promoting Ang II effects such as vasoconstriction, oxidative stress, and inflammation, all of which contribute to elevated blood pressure. However, in HFD-fed mice treated with a pressor dose of Ang II, plasma aldosterone levels showed an increasing tendency but remained statistically non-significant compared with ND-fed mice (ND: 102.8±11.6 pg/mL vs. HFD: 141.8±15.0 pg/mL, p=0.081), as shown in *Figure 7—figure supplement 3*, indicating a limited response in aldosterone production under these conditions. Additionally, BP did not significantly change (*Figure 7—figure supplement 2f, g*), potentially due to heterogeneous cellular responses across cell types, as indicated by the lack of reaction in vascular endothelial cells, vascular smooth muscle cells, and macrophages (*Figure 3—figure supplement 1*), and/or possibly due to aldosterone saturation from the high Ang II dose.

Finally, the current findings unequivocally demonstrated the molecular interactions between key molecules associated with dyslipidemia and hypertension in the kidneys. Moreover, this interaction can be effectively inhibited by ARBs, suggesting an additive effect in preventing the development of CKD, particularly in patients with hypertension and dyslipidemia. Interestingly, Ang II-dependent hypertensive animal models, including constriction of the renal artery and infusion of Ang II in rats and mice, have revealed a progressive increase in intrarenal Ang II levels, surpassing what can be accounted for by circulating Ang II levels alone (*Navar et al., 2011*). This is due to Ang II-dependent renal activation of the RAS, as indicated by increased urinary angiotensinogen (AGT) secretion (*Navar et al., 2011*; *Navar, 2013*). Importantly, the elevation of urinary AGT is also evident in patients with various pathologies, including hypertension and CKD, implying that renal Ang II levels increase even in individuals who do not exhibit elevated levels of circulating Ang II (*Kobori et al., 2009*; *Navar, 2013*; *Mills et al., 2012*). Taken together, the current finding of the synergistic effect of Ang II and oxLDL on AT1 activation in renal tissue is highly relevant for the development of kidney disease. RAS inhibitors, ARB, and ACE inhibitors are prioritized therapies to prevent the development of CKD with proteinuria in patients with hypertension (*Kidney Disease: Improving Global Outcomes Blood Pressure Work, G. KDIGO, 2021*). In addition to the well-established inhibitory effects of Ang II on the contraction of efferent arterioles (*Yang et al., 2011*), a novel renal protective action of RAS inhibitors has been proposed. Particularly in hypertension accompanied by dyslipidemia, RAS inhibitors may exhibit anti-inflammatory, antifibrotic, and antioxidant effects in the kidneys by inhibiting the Gq signaling pathway through the AT1-LOX-1 complex in renal tubular cells and fibroblasts. In terms of ARBs' effectiveness in inhibiting AT1/LOX-1 receptor conformational changes, all ARBs generally block the downstream signaling from AT1-LOX-1 interaction by preventing Ang II binding to AT1. Our previous study also showed that ARBs, including olmesartan, telmisartan, valsartan, and losartan, could inhibit

AT1 activation by oxLDL in the absence of Ang II (*Yamamoto et al., 2015*). However, certain ARBs—olmesartan, telmisartan, and valsartan—also act as inverse agonists, reducing baseline AT1 activity by preventing conformational changes even without Ang II (*Yamamoto et al., 2015*). This inverse agonist property may offer additional therapeutic benefits by reducing receptor activation beyond what is achieved by simple antagonism in pathological states where AT1 activation occurs independently of Ang II, such as in oxLDL presence. Collectively, the current findings suggest that RAS inhibitors, some of which possess inverse agonist properties, can concomitantly mitigate the effect of increased renal Ang II and oxLDL levels on the development of CKD in patients with hypertension and dyslipidemia, although direct evidence in clinical studies to support this remains to be elucidated.

This study has several limitations as follows: (1) The kidney, a complex organ vital for maintaining homeostasis, comprises a myriad of distinct cell types working in concert to execute its multifaceted functions (*Balzer et al., 2022*). The experiments conducted in mice raised questions regarding the specific cell types implicated in the synergistic effect of Ang II and oxLDL within the LOX-1-AT1 complex in the kidney. Addressing this issue would ideally require single-cell analysis, which is a challenge for future research. (2) The study employed systemic LOX-1 knockout mice. For a more detailed analysis, a phenotypic investigation using renal tissue-specific LOX-1 and AT1 knockout mice is required. Of particular importance is the analysis using tubule-specific knockout mice, in which the localization of these components has been verified through immunohistochemical staining. (3) The administration of Ang II (both pressor and subpressor doses) and its combination with HFD did not result in any histological changes in terms of fibrosis and mesangial expansion, despite the observed alterations in the associated gene expression and urinalysis for renal injury. Alterations in gene expression and urinalysis for renal injury may be sensitive and relatively early phenomena, and this discrepancy could potentially be attributed to the relatively short intervention duration of 4 wk for combined HFD and Ang II administration. Therefore, concurrent pathohistological alterations in the renal tissue might become evident with a more extended intervention period. (4) This study was limited by its inability to detect the amplification of Gq signaling in mouse renal tissue due to the concurrent administration of Ang II and HFD. Overcoming this limitation is a challenge for future studies.

In conclusion, the current findings suggest that the simultaneous binding of oxLDL and Ang II to their respective receptors within the complex induces a distinct conformational change compared

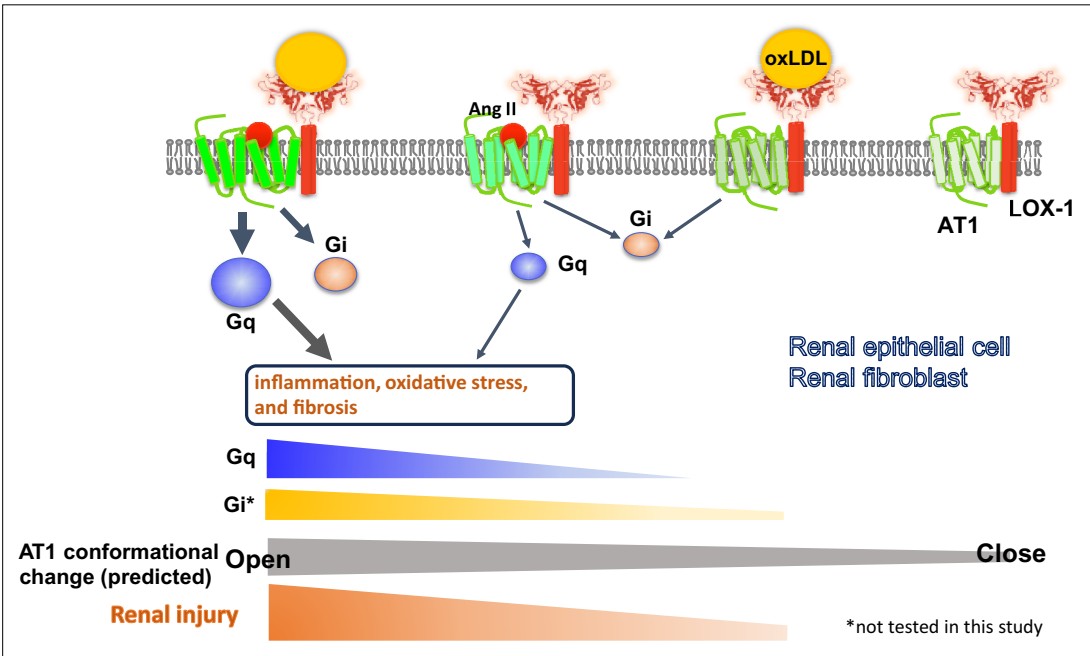

**Figure 10.** Schematic overview of the AT1 and LOX-1 Interaction dynamics in renal cells. This schematic summary illustrates the predicted structure-activation relationship of the AT1 receptor within the LOX-1-AT1 complex in renal component cells. This highlights how the simultaneous binding of angiotensin II (Ang II) to AT1 and oxidized low-density lipoprotein (oxLDL) to LOX-1 induces conformational changes in AT1. These changes were more pronounced than those triggered by the individual ligands. Such structural alterations have been proposed to amplify G protein αq subunit (Gq) signaling pathway activation, subsequently leading to renal damage.

with the effect of each ligand alone. This unique conformational change results in the heightened activation of G protein signaling and subsequent unfavorable cellular reactions in renal component cells. The relevance of this phenomenon was confirmed in mouse models, in which renal dysfunction was prominently exacerbated when there was a concomitant increase in Ang II and oxLDL levels. Notably, this effect was abolished by the deletion of LOX-1, indicating LOX-1 dependency of this in vivo phenomenon (*Figure 10*). These findings indicate the clinical relevance of the direct interaction between hypertension and dyslipidemia and further support the clinical significance of RA inhibition in treating patients with CKD.

## Materials and methods

### Cell culture and materials

HUVECs and BAECs were cultured in EGM-2 (Lonza, Basel, Switzerland). Cells with fewer than five passages were used in the experiments. Transgenic CHO cells were maintained in an F-12 Nutrient Mixture with GlutamaxTM-I (Thermo Fisher Scientific, MA, USA), 10% fetal bovine serum (FBS; Gibco, USA), and appropriate selection reagents, as described below. CHO-K1 cells were maintained in F-12 Nutrient Mixture with GlutamaxTM-I and 10% FBS. HAVSMCs were cultured in Dulbecco's modified Eagle's medium/F12 (DMEM/F12) (Nacalai Tesque, Japan) supplemented with 1% penicillin-streptomycin (Fujifilm, Japan) and 10% FBS. A10 cells were grown in DMEM (Wako, Osaka, Japan) supplemented with 10% FBS and 1% penicillin-streptomycin. NRK52E and NRK49F cells (ECACC, UK) were cultured in DMEM (Wako, Japan) supplemented with 5% FBS and the appropriate selection reagents. Gene transcription in CHO cells was induced by adding 100 ng/mL doxycycline (Merck KGaA, Darmstadt, Germany). Cells were incubated at 37°C in 5% $CO_2$ and 95% air. All the cell lines were tested negative for mycoplasma contamination.

### Construction of plasmid vectors

For stable transformants, pTRE2hyg vector (Clontech, USA) encoding mutated hAT1 with impaired ability to activate β-arrestin (pTRE2hyg-HA-FLAG-hAT1mg) were created using site direct mutagenesis as previously described (*Takahashi et al., 2021*). For real-time imaging, LOX-1 tagged with V5−6xHis at the C-terminus (V5-LOX-1) was subcloned into pmScarlet_C1 (plasmid #85042; Addgene) (mScarlet-LOX-1). HA-FLAG-hAT1 was subcloned into pcDNA3-EGFP (plasmid #85042; Addgene) (AT1-eGFP) (*Takahashi et al., 2021*).

### Stable transformants

We constructed CHO-K1 cells expressing tetracycline-inducible human LOX-1 tagged with V5−6xHis at the C-terminus (CHO-LOX-1), human HA-FLAG-hAT1 (CHO-AT1), or cells expressing both human LOX-1 and AT1 (CHO-LOX-1-AT1), as previously described (*Yamamoto et al., 2015*, *Takahashi et al., 2021*) To establish cells expressing both LOX-1 and mutated AT1 (pTRE2hyg-HA-FLAG-hAT1mg), they were co-transfected with the pSV2bsr vector (Funakoshi, Japan) into CHO-LOX-1 using the Lipofectamine 2000 transfection reagent (Thermo Fisher Scientific, USA). The stable transformants were selected with 400 µg/mL of hygromycin B (Wako, Osaka, Japan) and 10 µg/mL of blasticidin S (Funakoshi, Japan). The resistant clone expressing LOX-1 and mutated AT1 in response to doxycycline (Calbiochem, USA) was selected for use in the experiments (CHO-LOX-1-AT1mg) (*Takahashi et al., 2021*).

### Small interfering RNA

NRK52 and NRK47 cells were plated at 50% confluence on the day of transfection. Silencer Select small interfering RNA (siRNAs) for *Olr1* and *Agtr1a* (Thermo Fisher Scientific, MA, USA) were transfected into cells in a medium without serum or antibiotics using Lipofectamine RNAiMAX (Thermo Fisher Scientific, MA, USA), according to the manufacturer's instructions.

### Preparation of oxLDL

Human plasma LDL (1.019–1.063 g/mL), isolated by sequential ultracentrifugation, was oxidized using 20 µM $CuSO_4$ in PBS at 37 °C for 24 h. Oxidation was terminated by adding excess EDTA. LDL oxidation was analyzed by agarose gel electrophoresis for migration versus LDL (*Yamamoto et al., 2015*).

## Quantification of cellular IP1 accumulation

Gq-dependent activation of phospholipase C was quantified by measuring IP1 using the IP-One assay kit (Cisbio, France) as previously described (*Huang et al., 2022*). Briefly, cells were seeded at 80,000 cells/well in 96-well transparent cell culture plates and incubated under serum-free conditions for 24 hr. Thereafter, cells were treated for 1 hr with IP1 stimulation buffer, including vehicle, native LDL, oxLDL, Ang II, AGE, PTX (Merck KGaA, Darmstadt, Germany), YM-254890 (Fujifilm Wako, Osaka, Japan), and RKI-1448 (Selleck, USA), as described in the text. Cell lysates with Triton X at a final concentration of 1% were transferred to a 384-well white plate, and IP1 levels were measured by incubating the cell lysates with FRET reagents (cryptate-labeled anti-IP1 antibody and d2-labeled IP1 analog).

## Quantitative real-time PCR

RNA samples were purified using RNeasy Mini Kit (Qiagen, Germantown, MD, USA). One microgram of RNA was converted into cDNA using a ReverTra Ace qPCR RT kit (TOYOBO, Osaka, Japan) according to the manufacturer's instructions. All genes were evaluated using the ViiA7 Real-Time PCR System (Applied Biosystems, Thermo Fisher Scientific, Waltham, MA, USA). The data were analyzed using the ΔΔCt method with normalization against the GAPDH RNA expression in each sample. The primer sequences are listed in *Supplementary file 1*.

## Western blotting

Proteins were separated using sodium dodecyl sulfate-polyacrylamide gel electrophoresis and electrophoretically transferred onto polyvinylidene fluoride membranes. The membranes were blocked with 5% nonfat dried milk and incubated with primary antibodies overnight at 4 °C. The primary antibodies used in this study were as follows: anti-SMA antibody (1:1000), anti-α-Tubulin antibody (1:1000) (Cell Signaling Technology, Inc, Danvers, MA, USA). The bands were visualized with a chemiluminescence detection system (LAS-4000 mini; GE Healthcare Life Sciences, Buckinghamshire, UK) using Chemi-Lumi One Super (Nacalai Tesque, Kyoto, Japan).

## Calcium influx assay

Calcium influx was measured using Fura 2-AM (Dojindo, Kumamoto, Japan) with slight modifications to the manufacturer's protocol. In brief, cells plated in 96 wells were incubated with 5 µM Fura 2-AM in HEPES buffer saline (20 mM HEPES, 115 mM NaCl, 5.4 mM KCl, 0.8 mM MgCl2, 1.8 mM CaCl$_2$, 13.8 mM glucose, pH 7.4) for 1 hr at 37 °C, followed by replacement with recording medium without Fura 2-AM. Cells were treated with oxLDL, Ang II, or a combination of both at the indicated concentrations. Changes in F340/F380, an index of intracellular calcium concentration, were measured by dual-excitation microfluorometry using a digital image analyzer (Aquacosmos; Hamamatsu Photonics, Hamamatsu, Japan).

## BrdU assay

Proliferative activity was assessed using a BrdU Cell Proliferation ELISA Kit (Funakoshi, Japan). NRK49F cells were seeded in 96-well tissue culture plates and incubated with the test reagents for 24 hr. After incubating the cells with BrdU, we fixed the cells and denatured their DNA using a Fixing Solution. The plate was washed thrice with Wash Buffer before adding the Detector Antibody. Next, 100 µL/well of anti-BrdU monoclonal Detector Antibody was added, and the plate was incubated for 1 hr at room temperature. Subsequently, 100 µL/well of Goat Anti-Mouse IgG Conjugate was pipetted and incubated for 30 min at room temperature. After five washes, the reaction was stopped by adding Stop Solution to each well. The color of the positive wells changed from blue to bright yellow. Finally, the plate was read at a wavelength of 450/550 nm using a spectrophotometric microtiter plate reader.

## Creation of lentivirus encoding AT1 conformational sensors

For lentivirus encoding AT1 conformational sensors, rat AT1, and RlucII were subcloned into the pLVSIN-CMV Neo vector (Takara Bio, Japan). Next, the FlAsH binding sequence (CCPGCC) was inserted between residues K135 and S136 in the third intracellular loop (AT1-ICL3P3), as well as between residues K333 and M334 in the cytoplasmic-terminal tail (AT1-Ctail), utilizing the KOD-Plus Mutagenesis Kit (Toyobo, Japan), at the same site as previously documented (*Takahashi et al., 2021*).

## Intramolecular FlAsH Bioluminescence resonance energy transfer assay to detect AT1 conformational change

CHO-LOX-1 cells were initially plated onto a white clear-bottom 96-well culture plate at a density of $1 \times 10^5$ cells/well. The following day, the cells were transduced with lentivirus encoding AT1-Ctail or AT1-ICL3P3 (*Devost et al., 2017*) in 10% FBS. After 24 hr of transduction, the cultures were transferred to serum-free conditions and incubated for an additional 24 hr. Add 1.5 µM FlAsH-EDT$_2$ labeling reagent of TC-FlAsH II In-Cell Tetracysteine Tag Detection Kit (Thermo Fisher Scientific, MA, USA), washed twice with 250 µM BAL buffer, and assays were promptly conducted on a Spark microplate reader (TECAN, Switzerland). The BRET ratio (emission mVenus/emission Rluc) was calculated as follows: Following a 3 min baseline reading (with the final baseline reading presented at 0), cells were exposed to vehicle, oxLDL alone, AII alone, a combination of AII and oxLDL, or a combination of AngII, oxLDL, and LOX-1 antibodies. The BRET ratios were calculated every 16 s for a total of 320 s and the relative change in intramolecular BRET ratio was calculated by subtracting the average BRET ratio measured for cells stimulated with a vehicle at each time point.

## Analysis of LOX-1 and AT1 dynamics by real-time imaging

Live imaging was performed using previously reported methods (*Huang et al., 2022*). Briefly, 24 hr prior to imaging experiments, CHO-K1 cells were transfected with LOX-1-mScarlet and AT1-eGFP by electroporation. Subsequently, the cells were seeded in a 35 mm glass base dish (Iwaki, Japan) that had been pre-coated with a 1000 X diluted solution of 10 mg/mL poly-L-lysine (ScienCell, USA) 1 hr before seeding. The growth medium was substituted with imaging buffer (pH 7.4), which consisted of 125 mM NaCl, 5 mM KCl, 1.2 mM MgCl$_2$, 1.3 mM CaCl$_2$, 25 mM HEPES, and 3 mM D-glucose, with the pH adjusted to 7.4 using NaOH. Dynamic images of the cells were acquired at 25 °C using a SpinSR10 inverted spinning disk-type confocal super-resolution microscope (Olympus, Japan). The microscope was equipped with a 100 x NA1.49 objective lens (UAPON100XOTIRF, Olympus, Japan) and an ORCA-Flash 4.0 V2 scientific CMOS camera (Hamamatsu Photonics KK, Japan) at 5 s intervals. The imaging experiment was conducted using CellSens Dimension 1.11 software, employing a 3D deconvolution algorithm (Olympus, Japan), and the number of puncta was determined using ImageJ1.53K (*Huang et al., 2022*).

## Animals and diets

Male WT mice (C57BL/6 J) and LOX-1 KO mice with a C57BL/6 background were used in this study. LOX-1 KO mice were generated as described previously (*Mehta et al., 2007*). Mice were housed in a temperature-controlled (20–22°C) room on a 12 hr light/dark cycle and fed an ND (MF; Oriental Yeast, Osaka, Japan) or an HFD (High Fat Diet without DL-α-tocopherol, CLEA Japan Inc, Tokyo, Japan), which reported to increase plasma LOX-1 ligand in ApoE KO mice (*Kobori et al., 2009*). All study protocols were approved by the Animal Care and Use Committee of Osaka University (05-025-003) and were conducted according to the guidelines of the NIH for the Care and Use of Laboratory Animals.

## Blood pressure measurement in mice

The blood pressure of the mice was measured using the tail- cuff method with BP-98A (Softron, Japan). The measurements were performed after restraining the mice. The blood pressure was calculated as the average of 6 readings for each animal at each time point.

## Urine tests in mice

Urine tests in mice included the measurement of urine 8-OHdG, creatinine, and albumin concentrations. The DNA Damage (8-OHdG) ELISA Kit (StressMarq Bioscience, Canada), Creatinine Kit L type Wako (Fujifilm, Japan), and Mouse Albumin ELISA Kit (Bethyl Laboratories, Inc, TX, USA) were utilized for these measurements, following their respective instructions.

## Plasma LOX-1 ligand concentration

Measurement of LOX-1 ligands containing apoB (LAB) in mouse plasma was performed using a modified protocol based on a previously reported method (*Sato et al., 2008*). Briefly, recombinant human LOX-1 (0.25 µg/well) was immobilized on 384 well plates (Greiner, Frickenhausen, Germany)

by incubating overnight at 4 °C in 50 µl of PBS. After three washes with PBS, 80 µl of 20% (v/v) Immu-noBlock (KAC, Kyoto, Japan) was added, and the plates were incubated for 2 hr at 25 °C. After three washes with PBS, the plates were incubated for 2 hr at 25 °C with 40 µl of standard oxidized LDL or samples. Samples were prepared by fourfold dilution of plasma with HEPES-EDTA buffer (10 mM HEPES, 150 mM NaCl, 2 mM EDTA, pH 7.4), and standards were prepared by dilution of oxidized LDL with HEPES-EDTA buffer. Following three washes with PBS, the plates were incubated for 1 hr at 25 °C with chicken monoclonal anti-apoB antibody (HUC20, 0.5 µg/mL) in HEPES-EDTA containing 1% (w/v) BSA. After three washes with PBS, the plates were incubated for 1 hr at 25 °C with peroxidase-conjugated donkey anti-chicken IgY (Merck, NJ, USA) diluted 5000 times with HEPES-EDTA containing 1% (w/v) BSA. After five washes with PBS, the substrate solution containing 3,3',5,5'-tetramethylben-zidine (TMB solution, Bio-Rad Laboratories, CA, USA) was added to the plates and incubated them for 30 min at room temperature. The reaction was terminated with 2 M sulfuric acid. Peroxidase activity was determined by measuring absorbance at 450 nm using a SpectraMax 340PC384 Micro-plate Reader (Molecular Devices, CA, USA).

### Tissue preperation

Kidneys were perfused with cold PBS before removal. Kidney samples were rapidly excised. A quarter of samples were stored at 4 °C in RNAlater (Thermo Fisher Scientific, MA, USA) for RNA extraction. The remaining quarters were fixed in 4% paraformaldehyde overnight at 4 °C for histological evalu-ation. The remaining half was snap-frozen in liquid nitrogen and stored at –80 °C for further analysis.

### Periodic acid-Schiff and Masson-Trichrome staining

The degree of glomerular mesangial expansion and glomerular area (representing the structural integrity of the glomeruli) were assessed in a blinded manner using periodic acid-Schiff (PAS) staining. Collagen accumulation was determined by Masson-Trichrome (MTC) staining. For MTC staining, the area displaying fibrosis was quantitatively evaluated in a blinded manner by measuring the blue staining in six strongly magnified fields of view using the ImageJ software, and the average was calcu-lated after determining the ratio of the total area.

### Fluorescent immunostaining

For fluorescent immunostaining of LOX-1, AT-1, and megalin, the mice were perfused with cold saline before tissue removal. After 3 d of zinc fixation, the tissue was replaced with 70% ethanol. The 3-µm-thick kidney tissue sections were immunohistochemically stained with antibodies against ATGR (1:200, Cosmo Bio, Japan), OLR-1 (1:200, TS58 from the laboratory of T.S., Shinshu University School of Medicine, Nagano, Japan), and megalin (1:200, BiCell Scientific, MO, USA). Following deparaffin-ization (using Lemosol and gradient ethanol) and rehydration, the slices were subjected to antigen retrieval by autoclaving in citrate buffer (0.01 M; pH 6.0). Subsequently, the slices were washed thrice with PBS and blocked with 5% bovine serum albumin for 30 min at room temperature. The slides were then incubated with primary antibodies for 2 hr at room temperature. Goat anti-Rabbit IgG (H+L) High Cross-Adsorbed Secondary Antibody, Alexa Fluor Plus 488 and 594 (Thermo Fisher Scientific, MA, USA) were used as secondary antibodies for ATGR and OLR-1, respectively. After incubation with secondary antibodies for 1 hr at room temperature, the slices were washed with PBS. Finally, slides were sealed and photographed. Visual analyses were performed using a BZ-800L microscope (Keyence, Japan).

### Statistical analyses

All data are presented as the mean ± SEM. Differences between two treatments or among multiple treatments were determined using the Student's t-test or one-way ANOVA followed by Tukey's multiple comparison test.

## Acknowledgements

This work was partially supported by JSPS KAKENHI Grant Numbers 21K07389 (YT), 22K08181 (YN), 20H03576 (HR), and 18H02732 (KY). We are grateful to Tomoko Sato, Yoshinori Koishi, and Chika Takana for technical assistance. We would like to thank Editage (https://www.editage.com/) for the English language editing.

# Additional information

### Funding

| Funder | Grant reference number | Author |
|---|---|---|
| Japan Society for the Promotion of Science | 21K07389 | Yoichi Takami |
| Japan Society for the Promotion of Science | 22K08181 | Yoichi Nozato |
| Japan Society for the Promotion of Science | 20H03576 | Hiromi Rakugi |
| Japan Society for the Promotion of Science | 18H02732 | Koichi Yamamoto |

The funders had no role in study design, data collection and interpretation, or the decision to submit the work for publication.

### Author contributions

Jittoku Ihara, Conceptualization, Data curation, Formal analysis, Validation, Investigation, Visualization, Methodology, Writing – original draft, Writing – review and editing; Yibin Huang, Conceptualization, Data curation, Formal analysis, Validation, Investigation, Methodology, Writing – original draft; Yoichi Takami, Conceptualization, Data curation, Supervision, Funding acquisition, Validation, Methodology, Writing – original draft, Project administration, Writing – review and editing; Yoichi Nozato, Conceptualization, Data curation, Formal analysis, Funding acquisition, Validation, Methodology, Writing – original draft, Project administration, Writing – review and editing; Toshimasa Takahashi, Conceptualization, Resources, Data curation, Formal analysis, Investigation, Visualization, Methodology, Writing – original draft, Project administration, Writing – review and editing; Akemi Kakino, Resources, Data curation, Formal analysis, Investigation, Methodology, Writing – original draft; Cheng Wang, Data curation, Investigation; Ziwei Wang, Investigation; Yu Guo, Weidong Liu, Nanxiang Yin, Ryoichi Ohara, Taku Fujimoto, Shino Yoshida, Kazuhiro Hongyo, Hiroshi Koriyama, Hiroshi Akasaka, Hikari Takeshita, Formal analysis; Shinsuke Sakai, Supervision, Investigation; Kazunori Inoue, Supervision, Writing – review and editing; Yoshitaka Isaka, Supervision; Hiromi Rakugi, Conceptualization, Supervision, Funding acquisition, Writing – original draft; Tatsuya Sawamura, Conceptualization, Resources, Supervision, Writing – original draft, Writing – review and editing; Koichi Yamamoto, Conceptualization, Data curation, Formal analysis, Supervision, Funding acquisition, Validation, Investigation, Visualization, Writing – original draft, Project administration, Writing – review and editing

### Author ORCIDs

Yoichi Takami ⬥ https://orcid.org/0000-0001-9018-6707
Toshimasa Takahashi ⬥ https://orcid.org/0000-0002-5203-155X
Hiromi Rakugi ⬥ https://orcid.org/0000-0001-6508-4338

### Ethics

All study protocols were approved by the Animal Care and Use Committee of Osaka University (05-025-003) and were conducted according to the guidelines of the NIH for the Care and Use of Laboratory Animals.

Reviewer #1 (Public review): https://doi.org/10.7554/eLife.98766.3.sa1
Reviewer #2 (Public review): https://doi.org/10.7554/eLife.98766.3.sa2
Author response https://doi.org/10.7554/eLife.98766.3.sa3

# Additional files

### Supplementary files

Supplementary file 1. Primer sequences used in this study. This table lists the gene symbols, names, and primer sequences used in the study. The first section contains primer sequences for rats, while

the second section contains those for mice.
MDAR checklist

## Data availability
All data supporting the findings of this study are available within the paper and its supplementary information; source data are provided in this paper.

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
