## [Editor Report · eLife Assessment]

This study provides **useful** in vitro evidence to support a mechanism whereby dyslipidemia could accelerate renal functional decline through the activation of the AT1R/LOX1 complex by oxLDL and AngII. As such, it improves the knowledge regarding the complex interplay between dyslipidemia and renal disease and provides a **solid** basis for the discovery of novel therapeutic strategies for patients with lipid disorders. The methods, data, and analyses partly support the presented findings, although the observed variability and need for further in vivo validation require additional research in this key area.

---

## [Referee Report · Reviewer #1 (Public review)]

Summary:

In the present study, Dr. Ihara demonstrated a key role of oxLDL in enhancing Ang II-induced Gq signaling by promoting the AT1/LOX1 receptor complex formation.

Strengths:

This study is very exciting and the work is also very detailed, especially regarding the mechanism of LOX1-AT1 receptor interaction and its impact on oxidative stress, fibrosis and inflammation.

Weaknesses:

The direct evidence for the interaction between AT1 and LOX1 receptors in cell membrane localization is relatively weak.

---

## [Referee Report · Reviewer #2 (Public review)]

While the findings might be valid, there is enough uncertainty that these results should not be considered anything other than preliminary, warranting a more thorough and rigorous investigation.

Comments on revisions:

As the author mentioned that due to the receptor internalisation of AT1 and/or LOX1 induced by AngII or Ox-LDL makes it difficult to detect receptor interaction at the membrane by Co-IP. If so, the GPCR internalisation related pathway should be activated, such as GRKs, arrestin2 could be activated and enhanced during this process, whether they could further provide the evidence for these changes in different groups by Western blot or IF images.

If the authors don't know why the results across experiments can vary so greatly nor control them, how do we know that their interpretation of the very modest intra-experimental variability they observe is correct? They explain away the difference in biosensor activity response to the likely respective insertion sites that were used. While this can be true, and even might be true, it is important to note that the publication they cite shows that the sensors in the third loop and the C-terminus respond very similarly. In fact, the authors concluded: ‘Our results also suggest that positioning conformational biosensors into ICL3 and the C-tail effectively reports canonical G protein-mediated signaling downstream of the AT1R.’ Moreover, it is unclear why the less sensitive biosensor (as least as measured by degree of DBRET) is the one that appears to show enhancement. I suppose one could argue that the activity is maximal using the C-tail and one must use a less responsive reporter to detect the effect, but this is a rationalization for an unexplained result rather than a validated mechanistic explanation. If the other results were more compelling, perhaps this would be less of an issue. Finally, they did not explain why a control, non-specific antibody wasn't used for the studies presented in panel 2d. This would have been an easy study to have done in the interim. It also would have been important to test the effect of the LOX1-ab on the effects of AngII treatment alone.

In their response to the gene expression studies, the authors attribute the lack of a robust response for some genes to the low dose of oxLDL that was used but give no justification for their choice for this low dose. More importantly, they present the data for a number of hand-picked genes rather than a global assessment of response. Their justification---cost constraints---isn't sufficient to justify this incomplete analysis. Their selective rt-PCR results are a pilot study.

There is no direct evidence in this study that shows that ‘partial’ EMT is occurring in vivo. The rt-PCR studies presented in Fig 8 are not sufficient. Even if one accepts their incomplete analysis of transcriptomic studies using RT-PCR rather than a complete transcriptomic assessment, the study was done on bulk RNA from the entire kidney. The source material includes all cell types, not just epithelial cells, so there is no way to be sure that EMT is occurring. As noted elsewhere, they found no histologic evidence for injury and had no immunostaining results demonstrating ‘partial EMT’ of damaged renal epithelial cells.

All of the evidence described is indirect, and the responses, while plausible, are generally excuses for lack of truly unequivocally positive results. The authors acknowledge the potential confounders of lower BP response in the Lox1-KO, unexpected weight loss in response to high fat diet, the lack of meaningful histologic evidence of injury, and they also acknowledge the absence of increased Gq signaling in the kidney, which is central to their model, but defend the entire model based on some minor changes in urinary 8-OHdG and albumin levels and a curated set of transcriptional changes. Their data could support their model---loss of Lox1 seems to reduce the levels somewhat, but the data are preliminary.

There remain serious reservations about the immunostaining results, with explanations and new data not reassuring. The authors report that they are unable to co-stain for Lox1 and AT1R because both were generated in rabbit, but this reviewer didn't ask for co-staining of the two markers. Rather, it was co-staining showing that Lox1 and ATR1 in fact stain in a specific manner to the same nephron segments. The authors have added a supplementary figure showing co-staining for LOX1/AT1R with megalin, a marker for proximal tubules. However, several aspects of this are problematic:

i. The pattern in the new Supp Fig 10 does not look like that in Fig 9. In the latter, staining is virtually everywhere, all nephron segments, and predominantly basolateral. In Supp Fig 10, they note that the pattern is primarily in the microvilli of the proximal tubule, where megalin is present. The new studies also seem to be a bit more specific, ie there are some tubules that appear to not stain with the markers.

ii. It is difficult to be certain that the megalin staining isn't simply ‘bleed-through’ of the signal from the other antibody. The paper doesn't describe the secondary antibody used for megalin to be sure that the emission spectra completely non-overlapping and it isn't clear that the microscope that was used offers necessary precision.

iii. Their explanation for the pattern of AT1R staining is unconvincing. AT1R immunolocalization is known to be challenging, prompting Schrankl et al to do a definitive study using RNAscope to localize its expression in mice, rats and humans (Am J Physiol Renal Physiol 320: F644-F653, 2021). It argues against the pattern seen in Figure 9 (diffuse tubular expression), though it does suggest it is present in proximal tubules in mice. But perhaps more problematic for their model is that AT1R is not expressed in human tubules (or at least the RNA is undetectable).

Why isn't there more colocalization apparent for the AT1R and LOX1 if they form a co-receptor complex? They say that the complexes may be very dynamic, yet their movie in Suppl Fig 1 does not really support that. Not only are there few overlapping puncta in the static image, there is very little change over the duration of the movie. We don't see complexes form and then disappear and we see few new complexes form.

The explanation for why the number of replicates is variable is not reassuring. The authors note that it was because of the higher variability of the results, necessitating a higher ‘N’ to achieve significance, but this has the appearance of P-chasing.

---

## [Author Response]

The following is the authors’ response to the original reviews.

**Reviewer #1 (Public Review):**
Summary:This study demonstrates a key role of oxLDL in enhancing Ang II-induced Gq signaling by promoting the AT1/LOX1 receptor complex formation. Importantly, Gq-mediated calcium influx was only observed in LOX1 and AT1 both expressing cells, and AT1-LOX1 interaction aggravated renal damage and dysfunction under the condition of a high-fat diet with Ang II infusion, so this study indicated a new therapeutic potential of AT1-LOX1 receptor complex in CKD patients with dyslipidemia and hypertension.Strengths:This study is very exciting and the work is also very detailed, especially regarding the mechanism of LOX1-AT1 receptor interaction and its impact on oxidative stress, fibrosis, and inflammation.Weaknesses:The direct evidence for the interaction between AT1 and LOX1 receptors in cell membrane localization is relatively weak. Here I raise some questions that may further improve the study.Major points:(1) The authors hypothesized that in the interaction of AT1/LOX1 receptor complex in response to ox-LDL and AngII, there should be strong evidence of fluorescence detection of colocalization for these two membrane receptors, both in vivo and in vitro. Although the video evidence for AT1 internalization upon complex activation is shown in Figure S1, the more important evidence should be membrane interaction and enhanced signal of intracellular calcium influx.

Thank you for your valuable feedback. We agree that demonstrating the colocalization and interaction of AT1 and LOX-1 receptors at the membrane is critical to supporting our hypothesis.

In response, we have previously provided visual evidence of membrane co-localization of the AT1/LOX-1 receptor complex using an in situ PLA assay with anti-FLAG and antiV5 antibodies in CHO cells expressing FLAG-tagged AT1 and V5-tagged LOX-1 (Yamamoto et al., FASEB J 2015). This was further supported by immunoprecipitation of membrane proteins in CHO cells co-expressing LOX-1 and AT1, which confirmed the presence of the receptor complex. In the current study, we offer additional evidence of enhanced intracellular calcium influx following simultaneous stimulation with oxLDL and Ang II, confirming the functional activation of the AT1/LOX-1 receptor complex (Fig. 1g-j and Fig. 3e-h). Together, these findings provide substantial support for the colocalization of AT1 and LOX-1 and their influence on downstream signaling in our in vitro experiments.

However, we acknowledge the limitation of direct evidence for membrane co-localization of LOX-1 and AT1 in vivo. This constraint is attributed to the fact that both available anti-AT1 and anti-LOX-1 antibodies are derived from rabbits, making coimmunofluorescence or PLA challenging in our study. To address this, we employed coimmunofluorescent staining with megalin, a well-established marker for proximal renal tubules, as shown in Fig. S10. We found that both AT1 and LOX-1 co-localized with megalin, particularly at the brush borders, indicating their presence in the same renal compartments relevant to AT1/LOX-1 signaling.

We have revised the manuscript to highlight the functional evidence from calcium influx assays, supported by prior PLA results, demonstrating the interaction between LOX-1 and AT1. Additionally, we included a figure showing the co-localization of AT1 and LOX-1 with megalin in proximal renal tubules to reinforce these findings. Lastly, we have emphasized in the discussion the limitation regarding the lack of direct in vivo evidence for membrane co-localization of LOX-1 and AT1.

(2) Co-IP experiment should be provided to prove the AT1/LOX1 receptor interaction in response to ox-LDL and AngII in AT1 and LOX1 both expressing cells but not in AT1 only expressing cells.

We thank the reviewer for the insightful suggestion to validate the AT1/LOX1 receptor interaction under various stimulation conditions. In our previous study (Yamamoto et al., FASEB J 2015), we demonstrated the interaction between AT1 and LOX1 receptors through Co-IP and in situ PLA assays in cells overexpressing both receptors, without stimulation. These experiments provided solid evidence of the receptor interaction under static conditions at the cell membrane.

However, as noted in the previous work, we did not perform Co-IP experiments under AngII or oxLDL stimulation. The primary reason for this is that both AngII and oxLDL trigger internalization of the AT1 and/or LOX1 receptors, which may complicate the detection of receptor interaction at the membrane via Co-IP. This is supported by our realtime imaging, which showed a reduction in AT1 and/or LOX1 puncta following stimulation, indicating internalization of the receptors (Fig. 2a).

While we acknowledge the reviewer’s interest in investigating the interaction under AngII stimulation, we believe that the current data—especially from the PLA and Co-IP assays under static conditions—strongly support the interaction of AT1 and LOX1 receptors at the membrane.

(3) The authors mentioned that the Gq signaling-mediated calcium influx may change gene expression and cellular characteristics, including EMT and cell proliferation. They also provided evidence that oxidative stress, fibrosis, and inflammation were all enhanced after activating both receptors and inhibiting Gq was effective in reversing these changes. However, single stimulation with ox-LDL or AngII also has strong effects on ROS production, inflammation, and cell EMT, which has been extensively proved by previous studies. So, how to distinguish the biased effect of LOX1 or AT1r alone or the enhanced effect of receptor conformational changes mediated by their receptor interaction? Is there any better evidence to elucidate this point?

Thank you for raising this important point regarding the distinction between the individual effects of LOX-1 or AT1R activation and the enhanced effects mediated by their interaction. In our study, the concentration of oxLDL used (2–10 μg/ml) was significantly lower than concentrations typically employed in other studies (which often exceed 20 μg/ml). As a result, oxLDL alone produced minimal effects, aside from a reduction in cell proliferation observed in the BrdU assay. This suggests that oxLDL, at the concentrations used in our experiments, does not elicit a strong cellular response on its own.

The key to distinguishing the effect of the LOX-1/AT1 interaction lies in the amplification of Gq signaling, a pathway specifically activated by AngII. The distinction between the individual effects of LOX-1 or AT1R and the enhanced effects due to their interaction is centered on the increased activation of Gq signaling. In our experiments, co-treatment with oxLDL and AngII led to a significant increase in IP1 levels and calcium influx— both critical indicators of Gq signaling activation. While AngII alone also raised IP1 levels, the combined treatment with oxLDL further amplified the Gq signaling response, as reflected in the enhanced calcium influx. Importantly, oxLDL alone did not alter IP1 levels, even at high concentrations (100 μg/ml) (Takahashi et al., iScience 2021).

This enhancement of Gq signaling provides strong evidence of the synergistic interaction between LOX-1 and AT1, which surpasses the individual effects of either receptor alone. The LOX-1/AT1 interaction is thus crucial for the observed amplification of AngIIspecific signaling pathways. The combination of increased IP1 levels and calcium influx serves as compelling evidence of this interaction, clearly differentiating the effects of individual receptor activation from the enhanced response driven by receptor conformational changes and interaction.

Thank you again for your insightful comment, which has helped us to better articulate the significance of receptor interaction in this study.

(4) How does the interaction between AT1 and LOX1 affect the RAS system and blood pressure? What about the serum levels of rennin, angiotensin, and aldosterone in ND-fed or HFD-fed mice?

Thank you for your insightful question regarding the effects of AT1 and LOX-1 interaction on the renin-angiotensin system (RAS) and blood pressure, as well as the plasma levels of renin, angiotensin, and aldosterone in normal diet (ND)-fed and high-fat diet (HFD)-fed mice.

OxLDL binds to LOX-1, amplifying AT1 receptor activation and Gq signaling, which enhances the effects of Ang II. This interaction between AT1 and LOX-1 can lead to increased vasoconstriction, oxidative stress, and inflammation, which contribute to elevated blood pressure. This pathway may play a crucial role in modulating the RAS, particularly under conditions of elevated oxLDL, such as those induced by a HFD. Regarding the components of the RAS, we focused on plasma aldosterone levels, as this is a direct consequence of Ang II signaling. As shown in Fig. S7, when mice were treated with a pressor dose of Ang II infusion and subjected to a HFD to elevate oxLDL levels, we did not observe a significant increase in plasma aldosterone levels (102.8 ± 11.6pg/mL vs. 141.8 ± 15.0 pg/mL, P = 0.081).

In terms of blood pressure, Fig. 7b shows that no significant changes were observed under these treatment conditions, despite the AT1/LOX-1 interaction. These findings suggest that while oxLDL, via the AT1/LOX-1 interaction, can enhance Ang II signaling, its effect on blood pressure was not apparent in our study. This may be due to several factors, including heterogeneous cellular responses to the combined treatment across different cell types, as shown by the lack of reaction in vascular endothelial cells, vascular smooth muscle cells, and macrophages (Fig. S2). This may also be attributed to the high concentration of angiotensin II used in this study, which could have saturated aldosterone production under our experimental conditions. We have revised the manuscript to reflect these points.

Thank you again for your thoughtful comment, which has allowed us to expand and refine the discussion on this important aspect of our study.

**Reviewer #2 (Public Review):**
(1) Individuals with chronic kidney disease often have dyslipidemia, with the latter both a risk factor for atherosclerotic heart disease and a contributor to progressive kidney disease. Prior studies suggest that oxidized LDL (oxLDL) may cause renal injury through the activation of the LOX1 receptor. The authors had previously reported that LOX1 and AT1 interact to form a complex at the cell surface. In this study, the authors hypothesize that oxLDL, in the setting of angiotensin II, is responsible for driving renal injury by inducing a more pronounced conformational change of the AT1 receptor which results in enhanced Gq signaling.They go about testing the hypothesis in a set of three studies. In the first set, they engineered CHO cell lines to express AT1R alone, LOX1 in combination with AT1R, or LOX1 with an inactive form of AT1R and indirectly evaluated Gq activity using IP1 and calcium activity as read-outs. They assessed activity after treatment with AngII, oxLDL, or both in combination and found that treatment with both agents resulted in the greatest level of activity, which could be effectively blocked by a Gq inhibitor but not a Gi inhibitor nor a downstream Rho kinase inhibitor targeting G12/13 signaling. These results support their hypothesis, though variability in the level of activation was dramatically inconsistent from experiment to experiment, differing by as much as 20-fold. In contrast, within the experiment, differences between the AngII and AngII/oxLDL treatments, while nominally significant and consistent with their hypothesis, generally were only 10-20%. Another example of unexplained variability can be found in Figures 1g-1j. AngII, at a concentration of 10-12, has no effect on calcium flux in one set of studies (Figure 1g, h) yet has induced calcium activity to a level as great as AngII + oxLDL in another (Figure 1i). The inconsistency of results lessens confidence in the significance of these findings. In other studies with the LOX1-CHO line, they tested for conformational change by transducing AT1 biosensors previously shown to respond to AngII and found that one of them in fact showed enhanced BRET in the setting of oxLDL and AngII compared to AngII alone, which was blocked by an antibody to AT1R. The result is supportive of their conclusions. Limiting enthusiasm for these results is the fact that there isn't a good explanation as to why only 1 sensor showed a difference, and the study should have included a non-specific antibody to control for non-specific effects.

We sincerely appreciate the reviewer’s thorough and insightful feedback, especially regarding the variability observed in our experimental results. As the reviewer pointed out, the differences in activation levels between the calcium influx assay and the IP1 assay, particularly between AngII and AngII/oxLDL co-treatment, were indeed significant. These differences can be attributed to the inherent sensitivity of these assays, which are used to indirectly evaluate Gq activity. Despite the variability, we believe that the reliability of our results is supported by the consistent directional trends across both assays, which align with our hypothesis.

Regarding the inconsistencies in intracellular calcium dynamics observed in Fig. 1i, we have performed additional analysis of calcium kinetics during ligand stimulation, similar to the analysis in Fig. 1g. As shown in Author response image 1, the background signal in the experiment related to Fig. 1i was relatively higher than in Fig. 1g and 1h. This elevated background, which may have been influenced by variations between cells and experimental days, resulted in a higher percent change from baseline in samples treated with AngII alone. However, the combined effect of AngII with oxLDL was still apparent. This clarification further supports the consistency of our findings.

In reference to the BRET sensor experiments, we acknowledge the reviewer’s concern regarding the variability in sensor responses. As outlined in Devost et al. (J Biol Chem. 2017), the sensitivity of AT1 intramolecular FlAsH-BRET biosensors in detecting conformational changes induced by AngII is highly dependent on the insertion site of the FlAsH sequence. In our experiments, co-treatment with oxLDL and AngII enhanced AT1 conformational changes, but this effect was only detectable with the CHO-LOX-1-AT1-3p3 sensor (with FlAsH inserted in the third intracellular loop), and not with the CHO-LOX-1-AT1-C-tail P1 sensor (with FlAsH inserted at the C-terminal tail). This differential sensitivity likely explains why only one sensor showed a significant response, highlighting the critical role of FlAsH insertion site selection in these assays. We hope these clarifications address the reviewer’s concerns and improve confidence in the significance of our findings.

(2) The authors then repeated similar studies using publicly available rat kidney epithelial and fibroblast cell lines that have an endogenous expression of AT1R and LOX1. In these studies, oxLDL in combination with AngiI also enhanced Gq signaling, while knocking down either AT1R or LOX1, and treatment with inhibitors of Gq and AT1R blocked the effects. Like the prior set of studies, however, the effects are very modest and there was significant inter-experimental variability, reducing confidence in the significance of the findings. The authors then tested for evidence that the enhanced Gq signaling could result in renal injury by comparing qPCR results for target genes. While the results show some changes, their significance is difficult to assess. A more global assessment of gene expression patterns would have been more appropriate. In parallel with the transcriptional studies, they tested for evidence of epithelial-mesenchymal transition (EMT) using a single protein marker (alpha-smooth muscle actin) and found that its expression increased significantly in cells treated with oxLDL and AngII, which was blocked by inhibition of Gq inhibition and AT1R. While the data are sound, their significance is also unclear since EMT is a highly controversial cell culture phenomenon. Compelling in vivo studies have shown that most if not all fibroblasts in the kidney are derived from interstitial cells and not a product of EMT. In the last set of studies using these cell lines, the authors examined the effects of AngII and oxLDL on cell proliferation as assayed using BrdU. These results are puzzling---while the two agents together enhanced proliferation which was effectively blocked by an inhibitor to either AT1R or Gq, silencing of LOX1 had no effect.

Thank you for your thorough review and comments. We acknowledge your concerns regarding the modest effects observed and the variability in experimental outcomes. We would like to address your points systematically.

(1) Gq signaling and experimental variability:

Regarding the question of Gq signaling in Fig. 3, as previously mentioned, the observed differences in the IP1 assay are likely due to the sensitivity of the assay and the technical issues associated with detecting calcium influx and IP1 levels. While the overall differences between treatments may appear modest, the most critical comparison— between AngII alone and AngII combined with oxLDL—consistently showed significant differences, which aligns with the calcium influx results shown in Fig. 1. Notably, we found that the EC50 for IP1 production decreased by 80% in response to co-treatment with oxLDL and AngII, compared to AngII treatment alone. These findings demonstrate the robustness of Gq signaling enhancement with co-treatment, even if the absolute differences in the IP1 assay appear small.

(2) Gene expression in Fig. 4:

Regarding the gene expression analysis in Fig. 4, we used relatively low concentrations of oxLDL (5 μg/ml) compared to the higher concentrations typically employed in other studies (mostly exceeding 20 μg/ml). This may explain the lack of robust responses in some conditions. However, in combination with AngII, the co-treatment significantly upregulated several genes, particularly pro-inflammatory markers such as IL-6, TNFα, IL1β, and MCP-1 in NRK49F cells. These results suggest that the co-treatment induces a complex response, potentially activating multiple downstream signaling pathways beyond just Gq signaling, which may obscure more straightforward effects.

While we agree that a more global assessment of gene expression would provide further insights, due to cost constraints, we focused on key representative genes that are highly relevant to inflammation and fibrosis in this study.

(3) EMT in renal fibrosis:

We appreciate the reviewer’s insightful comments regarding the role of EMT in renal fibrosis. Regarding full EMT, in which epithelial cells completely transition into mesenchymal cells, previous studies using the unilateral ureteral obstruction (UUO) model suggest that full EMT may not play a significant role (J Clin Invest. 2011 Feb;121(2):468-74). The role of full EMT remains controversial in the context of renal fibrosis, with most kidney fibroblasts thought to originate from interstitial cells rather than through full EMT.

Recent studies, however, suggest that partial epithelial-mesenchymal transition (pEMT) could be involved in CKD, especially in association with inflammation, oxidative stress, and elevated TGF-β levels—conditions also present in our model involving Ang II infusion combined with an HFD. pEMT refers to a state in which epithelial cells acquire mesenchymal traits, such as increased α-SMA expression and secretion of pro-fibrotic cytokines, while remaining attached to the basement membrane without fully transitioning into fibroblasts (Front Physiol. 2020 Sep 15;11:569322). This phenomenon has been observed in kidney fibrosis models, including UUO, which shares inflammatory and oxidative stress conditions with our Ang II and HFD treatment model. The observed increase in α-SMA in our model may thus indicate a pEMT-like state, indirectly contributing to fibrosis through the secretion of growth factors and cytokines.

We are mindful of the importance of not overstating EMT's role. Accordingly, we interpret increased α-SMA expression as a potential marker of the pEMT process rather than definitive evidence of its presence or direct role in fibroblast formation. Furthermore, we acknowledge limitations in providing direct in vivo evidence for pEMT and recognize that further mechanistic studies are needed to elucidate its specific role in renal fibrosis, despite inherent challenges.

In response to the reviewer’s concern, we have revised the manuscript to clarify that our data support the possibility of pEMT contributing to fibrosis in this model, without overstating its impact. We also acknowledge the challenges in translating in vitro pEMT findings to in vivo models, where detecting the subtle effects of pEMT is inherently challenging.

(4) BrdU assay and fibroblast proliferation (Fig. 6b):

In Fig. 6b, the BrdU assay shows that fibroblast proliferation was significantly enhanced by the co-treatment with AngII and oxLDL, and this effect was abolished by LOX-1 knockdown, similar to the results observed with AT1 knockdown. These findings strongly suggest a combinatorial effect of AT1/LOX-1 interaction in promoting fibroblast proliferation, supporting the idea that the co-treatment operates through a coordinated mechanism involving both receptors. Notably, LOX-1 silencing did not affect the proliferation induced by AngII alone, as this response is independent of LOX-1.

We will incorporate these points into the Discussion section of the manuscript, specifically regarding the differences in sensitivity between the Ca influx and IP1 assays, as well as the emerging role of partial EMT in renal fibrosis. This will provide a clearer context for the interpretation of our findings and further strengthen the discussion on the significance of these phenomena.

Thank you again for your valuable feedback, which has helped us improve the clarity and depth of our manuscript.

(3) The final set of studies looked to test the hypothesis in mice by treating WT and Lox1KO mice with different doses of AngII and either a normal or high-fat diet (to induce oxLDL formation). The authors found that the combination of high dose AngII and a highfat diet (HFD) increased markers of renal injury (urinary 8-ohdg and urine albumin) in normal mice compared to mice treated with just AngII or HFD alone, which was blunted in Lox1-KO mice. These results are consistent with their hypothesis. However, there are other aspects of these studies that are either inconsistent or complicating factors that limit the strength of the conclusions. For example, Lox1- KO had no effect on renal injury marker expression in mice treated with low-dose AngII and HFD. It also should be noted that Lox1-KO mice had a lower BP response to AngII, which could have reduced renal injury independent of any effects mediated by the AT1R/LOX1 interaction. Another confounding factor was the significant effect the HFD diet had on body weight. While the groups did not differ based on AngII treatment status, the HFD consistently was associated with lower total body weight, which is unexplained. Next, the authors sought to find more direct evidence of renal injury using qPCR of candidate genes and renal histology. The transcriptional results are difficult to interpret; moreover, there were no significant histologic differences between groups. They conclude the study by showing the pattern of expression of LOX1 and AT1R in the kidney by immunofluorescence and conclude that the proteins overlap in renal tubules and are absent from the glomerulus. Unfortunately, they did not co-stain with any other markers to identify the specific cell types. However, these results are inconsistent with other studies that show AT1R is highly expressed in mesangial cells, renal interstitial cells, near the vascular pole, JG cells, and proximal tubules but generally absent from most other renal tubule segments.

Thank you for your valuable comments and for raising these important points. We appreciate the opportunity to clarify several aspects of our study and address the limitations and inconsistencies you have pointed out.

(1) Renal injury markers (urinary albumin and 8-OHdG) and the effect of LOX-1 loss of- function:

Our results showed that the combination of high-dose AngII and HFD led to a significant increase in renal injury markers, such as urinary albumin and 8-OHdG, in WT mice. In LOX-1 KO mice, this increase was significantly blunted, supporting a protective role of LOX-1 loss-of-function. However, as you noted, at low-dose AngII, there was no significant difference in urinary 8-OHdG between ND-fed and HFD-fed mice. Despite this, we observed a significant increase in urinary albumin in HFD-fed WT mice compared to ND-fed mice under low-dose AngII, and this difference was abolished in LOX-1 KO mice. Moreover, gene expression analysis showed that oxidative stress markers such as p67phox and p91phox (Fig. 8b), as well as p40phox, p47phox (Fig. S8), and inflammatory markers like IL1β (Fig. 8b), were significantly elevated in HFD-fed WT mice even with low-dose AngII, while these increases were absent in LOX-1 KO mice. These results suggest that the LOX-1/AT1 interaction contributes to renal injury under both low- and high-dose AngII conditions.

We acknowledge that the treatment duration may have influenced our results, as urine and renal tissue samples were only examined at a single time point (1.5 months after treatment initiation). The impact of AT1/LOX-1 interaction may evolve over time, and different treatment durations might yield varying outcomes. This is a limitation of our study, which we have addressed in the revised manuscript.

(2) Blood pressure and its effect on renal injury:

As shown in Fig. 7b and Fig S6f, LOX-1 KO mice exhibited a lower blood pressure response to high-dose AngII compared to WT mice, which could indeed have contributed to the reduced renal injury in the LOX-1 KO group, independent of the AT1/LOX-1 interaction. However, it is important to note that the differences in renal injury markers between AngII alone and AngII + HFD were largely abolished in LOX-1 KO mice, suggesting the in vivo relevance of the LOX-1/AT1 interaction observed in vitro. Additionally, as shown in Fig. 7d (urinary albumin), Fig. 8b (p67phox, p91phox), and Fig. S8b (p40phox, p47phox), even under subpressor doses of AngII, where no significant blood pressure differences were observed, HFD-fed WT mice exhibited exacerbated renal injury compared to ND-fed mice. These effects were ameliorated in LOX-1 KO mice, indicating that the protective effects in LOX-1 KO mice are at least partly independent of blood pressure changes and that the AT1/LOX-1 interaction plays a significant role in modulating renal injury under co-treatment with AngII and HFD.

(3) HFD and body weight changes:

We agree with your observation regarding the effect of HFD on body weight, which was consistently lower in HFD-fed groups, despite no differences in AngII treatment status. This is an atypical presentation compared to previous studies mostly showing increased body weight by feeding of HFD. The HFD used in this study was intended to elevate oxLDL levels, as previously reported (Atherosclerosis 200:303–309 (2008)). As shown in Fig. S6d and S6e, this can be attributed to reduced food intake in HFD-fed mice. Although modest, this weight reduction may influence renal function. This point is added in the limitation.

(4) Histological findings and qPCR results:

As discussed in the manuscript, despite significant changes in urinary markers and gene expression, we did not observe histological evidence of fibrosis or mesangial expansion, even under co-treatment with AngII and HFD. This may be due to the relatively short treatment period of 4 weeks, and a longer duration might be necessary to detect such changes. Additionally, we acknowledge that we did not detect increased Gq signaling in kidney tissue, which is another limitation of the study. Nevertheless, the gene expression data on oxidative stress, fibrosis, inflammation, and renal injury markers (e.g., p67phox, IL1β) are consistent with our hypothesis that the AT1/LOX-1 interaction exacerbates renal injury under AngII and HFD conditions.

(5) Immunostaining for AT1 and LOX-1:

Due to the use of rabbit-derived antibodies for both AT1 and LOX-1, it was technically not feasible to perform co-immunostaining for both receptors simultaneously. Instead, we performed co-immunofluorescent staining using megalin, a well-established marker of proximal renal tubules, to help localize these receptors. As shown in Fig. S10, both AT1 and LOX-1 were co-localized with megalin, particularly at the brush borders of proximal tubules. This pattern suggests the presence of these receptors in renal compartments relevant to AT1/LOX-1 signaling. While we did not perform additional co-staining with other markers to identify specific cell types, the strong localization with megalin provides robust evidence of their expression in proximal renal tubules, which is consistent with the literature on AT1R in this nephron segment. We acknowledge that previous studies have identified AT1R expression in mesangial cells, renal interstitial cells, the vascular pole, juxtaglomerular (JG) cells, and proximal tubules. In our immunofluorescence experiments, we did not detect significant AT1R expression in the glomerulus or mesangium. This finding aligns with other reports showing strong expression of AT1R in proximal tubules (Am J Physiol Renal Physiol. 2021 Apr 1;320(4)), although it does not exclude the possibility of AT1 expression in other compartments, given the sensitivity limitations of the immunofluorescence. Our focus on proximal tubules allowed us to observe clear AT1/LOX-1 co-localization in this region, particularly in the context of oxLDL and AngII signaling. Given that the AT1/LOX-1 interaction is crucial in kidney disease pathogenesis, this co-localization in proximal tubules highlights a key site of action for these receptors in the renal system.

In summary, while our study focused on the co-localization of AT1 and LOX-1 in proximal tubules, we agree that further exploration of AT1R expression in other renal cell types would provide a more comprehensive understanding of its role across different kidney compartments. We have addressed this in the revised discussion.

**Reviewer #1 (Recommendations For The Authors):**
Minor points:(1) In this study, AT1/LOX1 receptor complex was mainly observed in some renal cells, how about other types of cells that also highly express LOX1 and AT1r? Such as cardiomyocytes? Vascular endothelial cells?

Thank you for your insightful comment. In our study, we demonstrated that enhanced Gq signaling through co-treatment with AngII and oxLDL was not observed in other cell types, including vascular endothelial cells, smooth muscle cells, and macrophages, as indicated by the lack of an IP1 increase in response to the co-treatment (Fig. S2). The factors contributing to this heterogeneous response remain unclear, and further investigation is needed to explore this observation more thoroughly.

(2) Has the author detected such an effect on the AT2 receptor?

We greatly appreciate the reviewer’s insightful inquiry regarding the potential interaction between the AT2 receptor and LOX-1. In our previous work (Yamamoto et al., FASEB J 2015), we conducted an immunoprecipitation (IP) assay to investigate the interaction between LOX-1 and AT2 on cell membranes. The results of this assay demonstrated that, unlike AT1, LOX-1 exhibits minimal binding to the AT2 receptor under the experimental conditions tested. Specifically, our IP studies showed that while LOX-1 readily coimmunoprecipitated with AT1, indicating a strong interaction, this was not the case with AT2, where the binding was negligible. These findings suggest that the interaction between LOX-1 and AT1 is receptor-specific and that LOX-1 does not significantly associate with AT2 to influence signaling pathways.

(3) Which kind of ARBs are more effective for the inhibition of this AT1/LOX1 receptor conformational change?

Thank you for your insightful question regarding the effectiveness of ARBs in inhibiting the AT1/LOX-1 receptor conformational change. Based on our current understanding, any ARB should similarly block the downstream signaling resulting from the interaction between AT1 and LOX-1. This is because all ARBs function by inhibiting the binding of Ang II to AT1, thereby preventing receptor activation and the conformational changes that facilitate its interaction with LOX-1. Additionally, our previous study (FASEB J. 2015) demonstrated that even in the absence of Ang II, the activation of AT1 via the binding of oxLDL to LOX-1 was similarly blocked by ARBs, including olmesartan, telmisartan, valsartan, and losartan.

When oxLDL and Ang II are co-treated, the Gq signaling pathway is significantly amplified due to the interaction between LOX-1 and AT1. In this setting, all ARBs act by competitively inhibiting Ang II binding to AT1, effectively reducing Gq signaling.

However, a subtle but important difference arises when considering the inverse agonist activity of certain ARBs. Olmesartan, telmisartan, and valsartan are thought to act not only as competitive inhibitors of Ang II but also as inverse agonists, meaning they reduce the baseline activity of the AT1 receptor by preventing the conformational changes in the absence of Ang II. This inverse agonist property is particularly relevant in pathological conditions where AT1 receptor activation can occur independently of Ang II binding, such as in the presence of oxLDL. In these cases, ARBs with inverse agonist activity may offer an additional therapeutic advantage by reducing receptor activation beyond what is achieved by simple antagonism.

Thus, while the general efficacy of ARBs in blocking the AT1/LOX-1 interaction could be under similar conditions of oxLDL and Ang II co-treatment, ARBs with inverse agonist properties may provide additional benefit by further reducing AT1 activity.

We have revised the manuscript to clarify these points and to highlight the role of inverse agonist activity in ARB efficacy under these conditions.

Thank you again for your valuable comment, which has allowed us to refine our discussion on the relative efficacy of ARBs in inhibiting AT1/LOX-1 receptor interaction.

**Reviewer #2 (Recommendations For The Authors)**:My comments were pretty thorough in the public review. The only other comments I would add are the following:(1) Why are there so few overlapping LOX1 and ATR puncta in Supplementary Figure 1 if the receptors co-localize? The figure would suggest a very small proportion of the receptors actually are co-localized.

Thank you for your insightful comment regarding the apparent scarcity of overlapping LOX-1 and AT1R puncta in Fig. S1. We agree that at first glance, the low number of colocalized puncta may raise questions about the extent of interaction between these receptors. However, based on our previous findings reported in FASEB J 2015, we believe this phenomenon can be explained by the dynamic nature of the LOX-1 and AT1 interaction.

As we reported in FASEB J 2015, the interaction between LOX-1 and AT1 is sensitive to buffer conditions. Specifically, in non-reducing conditions, LOX-1 and AT1 form complexes, whereas in reducing buffer, this interaction is not observed. This suggests that the interaction between these receptors is not stabilized by strong covalent (disulfide) bonds but is instead transient, likely involving non-covalent interactions. Thus, LOX-1 and AT1 may form and dissociate repeatedly, contributing to a dynamic receptor complex rather than a permanent colocalization. This transient interaction could explain the relatively low number of overlapping puncta observed at a given time point in the liveimaging analysis.

Moreover, as you pointed out, it is likely that only a small fraction of LOX-1 and AT1 are physically co-localized at any one moment. However, when these receptors do interact, co-treatment with oxLDL and Ang II has been shown to significantly enhance Gq signaling. This suggests that the functional consequence of the LOX-1/AT1 interaction, particularly in response to stimuli such as oxLDL and Ang II, is more critical than the frequency of receptor colocalization at any one time.

We have revised the manuscript to include this explanation and to clarify the dynamic nature of the LOX-1/AT1 interaction. This revision also highlights the importance of considering not just the number of colocalized receptors but also the functional outcomes of their interaction, such as enhanced Gq signaling in response to co-treatment.

Thank you again for your careful observation, which has allowed us to better communicate the complexity of the receptor dynamics in our study.

(2) Tubulin is misspelled in Figure 5 (‘tublin’).

Thank you for pointing out the typographical error in Fig. 5. We have corrected the spelling of ‘tubulin’ in the revised figure. We appreciate your attention to detail, and we apologize for the oversight.

(3) Why does the number of replicates differ for some experimental sets i.e. Figure 1h vs other panels in Figure 1, Figure 2d vs other panels in Figure 2, Figure 7: Lox-1KO treated with High dose AngII and HFD? There aren't obvious reasons why the number of replicates should differ so much within a set of studies.

We are grateful to the reviewer for highlighting the discrepancies in the number of replicates across different figures in our manuscript. We would like to provide detailed explanations for each case.

(1) Fig. 1h vs Other Panels in Fig. 1:

The calcium influx assay (Fig. 1h) required a higher number of replicates due to the inherent biological variability associated with calcium signaling. To achieve statistical significance and account for variability in these measurements, we conducted additional replicates. Other panels, such as those measuring IP1 accumulation (Fig. 1a–f), displayed more consistent and reproducible results, allowing us to use fewer replicates while still maintaining statistical power.

(2) Fig. 2d vs Fig. 2b and 2c:

The difference in the number of replicates between Fig. 2d (N=8) and Fig. 2b and 2c (N=4) is due to the distinct nature of the measurements and the variability expected in each assay. In Fig. 2d, which measures the effects of a LOX-1 neutralizing antibody on BRET, additional replicates were needed to ensure the robustness of the statistical analysis due to the greater complexity and sensitivity of the assay. The inclusion of an antibody treatment introduces more variability, necessitating a higher number of replicates (N=8) to confidently assess the effects of the neutralizing antibody. In contrast, Fig. 2b and 2c involved BRET measurements of AT1 conformational changes without antibody intervention. These assays are more reproducible and have less experimental variability, allowing for a smaller sample size (N=4) while still achieving reliable and statistically significant results. The differences in sample size across these panels were carefully considered to ensure appropriate statistical power for each specific experimental condition.

(3) Fig. 7: LOX-1 KO Mice Treated with High-dose AngII vs Saline:

We acknowledge the reviewer’s concern regarding the higher number of LOX-1 KO mice treated with high-dose Ang II compared to the saline group. The number of saline-treated mice was indeed sufficient for reliable statistical analysis. However, the decision to increase the number of mice in the high-dose Ang II group was driven by the anticipated higher variability in the physiological responses under these conditions, such as blood pressure and renal injury. To ensure that we captured the full spectrum of responses and to maintain robust statistical power in the high-dose group, we opted to include more mice in this cohort.

We hope this response provides clarity on the rationale behind the varying number of replicates across different experiments. We have rigorously applied appropriate statistical methods to account for these differences, ensuring that the conclusions drawn are robust and scientifically sound. We appreciate the reviewer’s understanding of the experimental constraints and variations that can arise in complex studies such as these.